# Pathogen infection and cholesterol deficiency activate the *C. elegans* p38 immune pathway through a TIR-1/SARM1 phase transition

Nicholas D Peterson[1†], Janneke D Icso[2†], J Elizabeth Salisbury[1], Tomás Rodríguez[3], Paul R Thompson[2]*, Read Pukkila-Worley[1]*

[1]Program in Innate Immunity, Division of Infectious Diseases and Immunology, University of Massachusetts Chan Medical School, Worcester, United States; [2]Program in Chemical Biology, University of Massachusetts Chan Medical School, Worcester, United States; [3]RNA Therapeutics Institute, University of Massachusetts Chan Medical School, Worcester, United States

**\*For correspondence:**
Paul.Thompson@umassmed.edu
(PRT);
Read.Pukkila-Worley@
umassmed.edu (RP-W)

[†]These authors contributed equally to this work

**Competing interest:** The authors declare that no competing interests exist.

## Abstract

Intracellular signaling regulators can be concentrated into membrane-free, higher ordered protein assemblies to initiate protective responses during stress — a process known as phase transition. Here, we show that a phase transition of the *Caenorhabditis elegans* Toll/interleukin-1 receptor domain protein (TIR-1), an NAD$^+$ glycohydrolase homologous to mammalian sterile alpha and TIR motif-containing 1 (SARM1), underlies p38 PMK-1 immune pathway activation in *C. elegans* intestinal epithelial cells. Through visualization of fluorescently labeled TIR-1/SARM1 protein, we demonstrate that physiologic stresses, both pathogen and non-pathogen, induce multimerization of TIR-1/SARM1 into visible puncta within intestinal epithelial cells. In vitro enzyme kinetic analyses revealed that, like mammalian SARM1, the NAD$^+$ glycohydrolase activity of *C. elegans* TIR-1 is dramatically potentiated by protein oligomerization and a phase transition. Accordingly, *C. elegans* with genetic mutations that specifically block either multimerization or the NAD$^+$ glycohydrolase activity of TIR-1/SARM1 fail to induce p38 PMK phosphorylation, are unable to increase immune effector expression, and are dramatically susceptible to bacterial infection. Finally, we demonstrate that a loss-of-function mutation in *nhr-8*, which alters cholesterol metabolism and is used to study conditions of sterol deficiency, causes TIR-1/SARM1 to oligomerize into puncta in intestinal epithelial cells. Cholesterol scarcity increases p38 PMK-1 phosphorylation, primes immune effector induction in a manner that requires TIR-1/SARM1 oligomerization and its intrinsic NAD$^+$ glycohydrolase activity, and reduces pathogen accumulation in the intestine during a subsequent infection. These data reveal a new adaptive response that allows a metazoan host to anticipate pathogen threats during cholesterol deprivation, a time of relative susceptibility to infection. Thus, a phase transition of TIR-1/SARM1 as a prerequisite for its NAD$^+$ glycohydrolase activity is strongly conserved across millions of years of evolution and is essential for diverse physiological processes in multiple cell types.

## Editor's evaluation

Your manuscript makes an important contribution to delineating mechanisms involved in the activation of innate immune function in *C. elegans*. The reviewers as well as the editors find this study to be well-conducted, presenting a large amount of new and convincing data on the phase transition underlying p38 immune pathway activation, especially on the novel role of cholesterol metabolism in this process.

**eLife digest** From worms to humans, animals have developed various strategies – including immune defences – to shield themselves from disease-causing microbes. A type of roundworm, called *C. elegans*, lives in environments rich in microbes, so it needs effective immune defences to protect itself. The roundworms share a key regulatory pathway with mammals that helps to control their immune responses. This so-called p38 pathway relies on proteins that interact with each other to activate protective immune defences.

Proteins contain different regions or domains that can give them a certain function. For example, proteins with a region called TIR play important roles in immune defences in both animals and plants. One such protein, called SARM1, is unique among animal and plant proteins in that it is an enzyme, which cleaves an important metabolite in the cell. In *C. elegans*, the SARM1 homolog, TIR-1, controls the p38 pathway during infection, but how TIR-1 activates it is unclear.

To find out more, Peterson, Icso et al. modified *C. elegans* to generate a fluorescent form of TIR-1 and infected the worms with bacteria. Imaging techniques revealed that infection caused TIR-1 in gut cells to cluster into organized structures, which increases the enzymatic activity of the protein to activate the p38 immune pathway. Moreover, stress situations, such as cholesterol nutrient withdrawal, activated the p38 pathway in the same way. This adaptive stress response allows the animal to defend itself against pathogen threats during times, when they are most susceptible to infections.

Cells in the gut provide a primary line of defence against infectious bacteria and are important for maintaining a healthy gut immune system. When the mechanisms for pathogen sensing and immune maintenance are disrupted, it can lead to inflammation and higher risk of infection. Peterson, Icso et al. show how a key regulator of gut immunity, TIR-1, provides protection in *C. elegans*, which may suggest that SARM1 could have a similar role in mammals.

## Introduction

The p38 mitogen-activated protein kinase (MAPK) pathway is a key regulator of stress responses and innate immune defenses in metazoans. The *C. elegans* p38 homolog PMK-1 is part of a classic MAPK signaling cascade that is activated by the MAPKKK NSY-1 and MAPKK SEK-1, which are the nematode homologs of mammalian ASK and MKK3/6, respectively (*Kim et al., 2002*). As in mammals, the *C. elegans* p38 PMK-1 pathway regulates the expression of secreted innate immune effectors and is required for survival during pathogen infection (*Kim et al., 2002*; *Troemel et al., 2006*; *Peterson and Pukkila-Worley, 2018*; *Pukkila-Worley and Ausubel, 2012*). However, the mechanisms that activate the NSY-1/SEK-1/p38 PMK-1 signaling cassette in *C. elegans* intestinal epithelial cells are poorly defined.

Toll/interleukin-1 receptor (TIR) domain-containing proteins serve essential functions in both animal and plant immunity (*Zhang et al., 2017*; *Ve et al., 2015*; *O'Neill and Bowie, 2007*). In mammals, Toll-like receptors (TLRs) and downstream adaptor proteins contain TIR domains, which transduce intracellular signals upon pathway activation (*Gay et al., 2014*). Nucleotide-binding leucine-rich repeat (NLR) proteins in plants also contain TIR domains, which activate host defenses following recognition of pathogen-derived effector proteins (*Lolle et al., 2020*). In *C. elegans*, TIR-1, the homolog of mammalian sterile alpha and TIR motif-containing 1 (SARM1), acts upstream of NSY-1/ASK to control p38 PMK-1 activation (*Liberati et al., 2004*; *Couillault et al., 2004*). Interestingly, TIR-1/SARM1 is unique among animal and plant TIR domain-containing proteins in that it is an enzyme, which cleaves nicotinamide adenine dinucleotide (NAD$^+$) (*Essuman et al., 2017*; *Wan et al., 2019*). The NAD$^+$ glycohydrolase activity of TIR is activated upon oligomerization into a multimeric protein complex (*Horsefield et al., 2019*). NAD$^+$ depletion in this manner promotes cell death in plants during pathogen infection and triggers axonal degeneration following neuronal injury (*Gerdts et al., 2015*; *Gerdts et al., 2013*). However, it is not known whether the NAD$^+$ glycohydrolase activity of TIR is also required for immune function in metazoans.

Intracellular signaling regulators can be compartmentalized into membrane-less, higher-ordered protein assemblies with liquid droplet-like properties (*Du and Chen, 2018*; *Chen et al., 2020*; *Alberti et al., 2019*; *Boeynaems et al., 2018*; *Alberti and Dormann, 2019*; *Patel et al., 2015*; *Verdile et al., 2019*). Cytoplasmic de-mixing or phase transition of proteins in this manner concentrates signaling

regulators to facilitate rapid and specific activation of protective defenses during stress (*Du and Chen, 2018*; *Chen et al., 2020*; *Kroschwald et al., 2015*). Interestingly, human SARM1 oligomerizes and undergoes a phase transition, which enhances its intrinsic NAD$^+$ glycohydrolase activity (*Loring et al., 2021*). Induced multimerization of *C. elegans* TIR-1/SARM1 in neurons also correlated with enhanced axonal degeneration (*Loring et al., 2021*). Here, we show that a phase transition and NAD$^+$ glycohydrolase activity of TIR-1/SARM1 is required for p38 PMK-1 immune pathway activation and pathogen resistance in *C. elegans*. We labeled the TIR-1/SARM1 protein with a fluorescent tag at its genomic locus and demonstrated that TIR-1/SARM1 forms visible puncta within the intestine in response to physiologic stimuli, including both pathogen infection and cholesterol deficiency. By promoting multimerization of TIR-1/SARM1 with macromolecular crowding agents in vitro, we demonstrate that a phase transition is required for the catalytic activity of TIR-1/SARM1. Accordingly, TIR-1/SARM1 containing mutations that either specifically prevent the phase transition or impair NAD$^+$ hydrolysis shows decreased enzymatic activity in vitro. *C. elegans* carrying these same mutations, edited into the genome using CRISPR/Cas9, fail to induce p38 PMK phosphorylation, are unable to upregulate immune effector expression, and have enhanced susceptibility to bacterial infection.

We also report that a loss-of-function mutation in *nhr-8*, which alters cholesterol metabolism and is used to study conditions of sterol deficiency (*Magner et al., 2013*), induces TIR-1/SARM1 oligomerization and p38 PMK-1 pathway activation. *C. elegans* lacks the ability to synthesize cholesterol de novo and must acquire dietary sterols from the environment to support multiple aspects of cellular physiology, including development, fecundity, lifespan, and resistance against pathogen infection (*Hieb and Rothstein, 1968*; *Chitwood, 1999*; *Shim et al., 2002*; *Merris et al., 2003*; *Yochem et al., 1999*; *Otarigho and Aballay, 2020*; *Cheong et al., 2011*). Some, but not all, *C. elegans* larvae that encounter sterol-scarce environments enter an alternative developmental program, called dauer diapause, to promote animal survival (*Burnell et al., 2005*; *Vanfleteren and De Vreese, 1996*; *Gerisch et al., 2001*; *Gerisch and Antebi, 2004*; *Motola et al., 2006*). Here, we show that *C. elegans* larvae that do not enter dauer diapause in an environment devoid of dietary sterols adapt by promoting oligomerization of TIR-1/SARM1 in vivo to activate the p38 PMK-1 innate immune pathway through its intrinsic NAD$^+$ glycohydrolase activity. Priming p38 pathway activation in this manner augments immune effector expression during a subsequent bacterial infection and reduces pathogen accumulation in the intestine. Thus, we propose that activation of the p38 PMK-1 pathway during conditions of low cholesterol availability is an adaptive response to preempt pathogen attack during a time of relative vulnerability to infection.

## Results

### Multimerization of TIR-1/SARM1 and its intrinsic NAD$^+$ glycohydrolase activity are required for activation of the p38 PMK-1 innate immune pathway during pathogen infection

To determine if *C. elegans* TIR-1/SARM1 multimerizes to activate the p38 PMK-1 innate immune pathway, we used CRISPR/Cas9 to insert the fluorescent protein wrmScarlet at the C-terminus of the endogenous *C. elegans tir-1* locus, which labeled all *tir-1* isoforms. In uninfected animals, TIR-1::wrmScarlet is barely detectable in intestinal epithelial cells (*Figure 1A*). However, *P. aeruginosa* infection caused TIR-1::wrmScarlet to multimerize into visible puncta within intestinal epithelial cells (*Figure 1A and B*). We distinguished TIR-1::wrmScarlet puncta from autofluorescent gut granules by comparing images in the red and green fluorescence channels. TIR-1::wrmScarlet puncta are those that are seen in the red, but not the green fluorescence channel (arrowheads in *Figure 1A*), as opposed to gut granules, which can be seen in both channels (asterisks in *Figure 1A*).

*C. elegans* TIR-1 protein has three characterized domains: a Heat/Armadillo repeat domain, a sterile alpha motif (SAM) domain, and a Toll-interleukin receptor (TIR) domain (*Chuang and Bargmann, 2005*; *Figure 1C*). *C. elegans* TIR-1 oligomerizes in vitro through interactions of its SAM domains (*Horsefield et al., 2019*). We used CRISPR-Cas9 to delete both SAM domains in *tir-1* (*tir-1$^{\Delta SAM}$*) and to generate point mutants in two residues within the *C. elegans* TIR domain that are important for the self-association and activity of mammalian SARM1 (*C. elegans tir-1$^{G747P}$* and *tir-1$^{H833A}$*) (*Horsefield et al., 2019*; *Figure 1D*). The *C. elegans tir-1$^{\Delta SAM}$*, *tir-1$^{G747P}$* and *tir-1$^{H833A}$* mutants prevented activation of the p38 PMK-1-dependent immune reporter T24B8.5p::*gfp* in animals infected with *P. aeruginosa*



**Figure 1.** Multimerization of TIR-1/SARM1 and its intrinsic NAD⁺ glycohydrolase activity are required for activation of the p38 PMK-1 innate immune pathway during pathogen infection. (**A**) Images of animals expressing TIR-1::wrmScarlet in the indicated conditions. All *tir-1::wrmScarlet* animals were treated with *glo-3(RNAi)* to deplete autofluorescent gut granules. Representative images for each condition are displayed. Red fluorescent channel images display both TIR-1::wrmScarlet fluorescence and autofluorescent signal, while the green fluorescent channel images only display signals from autofluorescent gut granules. TIR-1::wrmScarlet puncta are indicated by arrowheads and autofluorescent gut granules by asterisks. Scale bar equals 20 μm (2 μm for the inset enlarged images). (**B**) The number of puncta present in the last posterior pair of intestinal epithelial cells in the red (*tir-1::wrmScarlet*), but not the green (autofluorescence) fluorescent channels were quantified using Fiji image analysis software. Each data point is the

*Figure 1 continued on next page*

Figure 1 continued

number of TIR-1::wrmScarlet puncta from one animal. The n is indicated for each condition. *equals p < 0.05 (two-way ANOVA with Tukey's multiple comparison testing). (C) Model of *tir-1* showing the domains and the mutations that were introduced using CRISPR-Cas9. (D) Expression of the innate immune effector T24B8.5p::*gfp* in *tir-1* mutants with predicted defects in oligomerization (*tir-1*$^{ΔSAM}$, *tir-1*$^{G747P}$ and *tir-1*$^{H833A}$) and NADase catalytic activity (*tir-1*$^{E788A}$) during *P. aeruginosa* infection. Scale bar equals 200 μm. (E) Immunoblot analysis of lysates from the indicated genotypes probed with antibodies targeting the doubly phosphorylated TGY epitope in phosphorylated PMK-1 (phos-PMK-1), total PMK-1 protein (total PMK-1), and tubulin (α-tubulin). *nsy-1(ag3)* and *pmk-1(km25)* loss-of-function mutants are the controls, which confirm the specificity of the phospho-PMK-1 probing. (F) The band intensities of three biological replicates of the Western blot shown in (E) were quantified. Error bars reflect SEM. *equals p < 0.05 (one-way ANOVA with Dunnett multiple comparison testing). (G) *C. elegans* pathogenesis assay with *P. aeruginosa* and *C. elegans* of indicated genotypes at the L4 larval stage are shown. Data are representative of three trials. Difference between wild-type and all *tir-1* mutants is significant (p < 0.05). The Kaplan-Meier method was used to estimate the survival curves for each group, and the log-rank test was used for all statistical comparisons. Sample sizes, mean lifespan and p-values for all trials are shown in **Supplementary file 4**. See also **Figure 1—figure supplement 1**.

The online version of this article includes the following source data and figure supplement(s) for figure 1:

**Source data 1.** *Figure 1B* Quantification of the number of TIR-1::wrmScarlet puncta present in the last posterior pair of intestinal epithelial cells.

**Source data 2.** *Figure 1F* Quantification of p38 immunoblot analysis of lysates from the indicated genotypes.

**Figure supplement 1.** Multimerization of TIR-1/SARM1 and its intrinsic NAD + glycohydrolase activity are required for activation of the p38 PMK-1 innate immune pathway during pathogen infection.

(*Figure 1D*). Consistent with these data, the *tir-1*$^{ΔSAM}$, *tir-1*$^{G747P}$ and *tir-1*$^{H833A}$ mutants have reduced levels of active, phosphorylated p38 PMK-1, equivalent to the *tir-1(qd4)* null allele (*Shivers et al., 2009*; *Figure 1E and F*). Additionally, these mutants are each markedly hypersusceptible to *P. aeruginosa* infection (*Figure 1G*).

The TIR domain of *C. elegans*, TIR-1, and its mammalian homolog, SARM1, possess intrinsic NADase activity (*Essuman et al., 2017*; *Horsefield et al., 2019*; *Summers et al., 2016*). Importantly, oligomerization of mammalian SARM1 and *C. elegans* TIR-1 is required for maximal NADase activity in vitro (*Horsefield et al., 2019*). The NADase activity in the TIR domain of mammalian SARM1 requires a putative catalytic glutamate residue (*Essuman et al., 2017*). We used CRISPR-Cas9 to mutate the homologous glutamate in *C. elegans tir-1* (*tir-1*$^{E788A}$) and found that it was required for the immunostimulatory activity of *tir-1* – *tir-1*$^{E788A}$ mutants do not induce T24B8.5p::*gfp* following *P. aeruginosa* infection (*Figure 1D*), had less active, phosphorylated p38 PMK-1 (*Figure 1E and F*), and were more susceptible to *P. aeruginosa* infection (*Figure 1G*). We confirmed that *tir-1*$^{E788A}$, *tir-1*$^{ΔSAM}$ and *tir-1*$^{G747P}$ mutants are translated and not degraded by introducing a 3xFLAG tag at the C-terminus of each mutant using CRISPR-Cas9 (*Figure 1—figure supplement 1A and B*) and probing for epitope-tagged mutant protein in western blots with an anti-FLAG antibody (*Figure 1—figure supplement 1A*). In addition, the 3xFLAG-tagged wild-type TIR-1 expressed T24B8.5p::gfp, but the tagged mutant TIR-1 proteins did not (*Figure 1—figure supplement 1B*). Collectively, these data demonstrate that multimerization of TIR-1 and its intrinsic NADase activity are required to activate the p38 PMK-1 innate immune pathway in the intestine during pathogen infection.

## TIR multimerization and phase transition superactivates its intrinsic NAD$^+$ glycohydrolase activity

To further characterize the mechanism of TIR-1/SARM1 activation, we recombinantly expressed and purified the TIR domain of the TIR-1 protein (called TIR) from *E. coli* and evaluated its NADase activity in vitro using an etheno-NAD$^+$ (ε-NAD) activity assay, in which hydrolysis of the nicotinamide moiety of ε-NAD leads to an increase in fluorescence. Interestingly, purified TIR only shows very modest NADase activity even at high protein concentrations ( > 15 μM) (*Figure 2A*). Notably, the NADase activity of TIR increased parabolically with increasing TIR concentrations rather than linearly, suggesting that multimerization of TIR-1 potentiates its NADase activity (*Figure 2A*).

Given that high concentrations of TIR are required to observe NADase activity, we hypothesized that molecular crowding might activate the enzyme. Therefore, we assessed the effect of several macro- and microviscogens on TIR activity. Macroviscogens reduce the free volume available for protein movement and thus promote aggregation of protein complexes that are capable of self-association (*Gadda and Sobrado, 2018*; *Blacklow et al., 1988*). Importantly, macroviscogens have minimal impact on the rate of diffusion of small molecules. By contrast, microviscogens, which are much smaller than most enzymes, affect the diffusion of substrates in solution and, thus, the frequency

**Figure 2.** TIR multimerization and phase transition superactivates its intrinsic NAD $^+$ glycohydrolase activity. (**A**) NADase activity of purified TIR at increasing TIR protein concentrations is shown. Activity was assessed by incubating TIR protein with 1 mM ε-NAD and monitoring the rate at which the fluorescent product ε-ADPR was produced. Curve represents a nonlinear regression fit of the NADase activity data points (n = 2). (**B**) NADase activity of 2.5 μM TIR incubated in the presence of 25% (w/v) of macro- (PEG 8000, PEG 3350, and dextran) and micro- (sucrose and glycerol) viscogens was

*Figure 2 continued on next page*

*Figure 2 continued*

assessed as described in A (n = 2). (**C**) Dose dependency of macroviscogens on the NADase activity of TIR is shown. A total of 2.5 μM TIR protein was incubated with the indicated PEG compounds at concentrations from 0% to 30% (w/v). NADase activity was assessed as described in A (n = 2). (**D**) Steady-state kinetic analysis of 2.5 μM TIR incubated in 0–30% (w/v) of PEG 3350 with the ε-NAD substrate at concentrations from 0 to 4000 μM was assessed as described in A. (n = 2). From the steady-state kinetic analysis performed in D, $K_m$(**E**), $k_{cat}$ (**F**), and $k_{cat}/K_m$ (**G**) were determined at each PEG 3350 concentration. (**H**) SDS-PAGE analysis of TIR protein fractions incubated with increasing concentrations of PEG 3350 precentrifugation (**C**) and after centrifugation, the soluble (**S**) and pellet (**P**) protein fractions. NADase activity of TIR protein in each fraction and at each concentration of PEG 3350 was assessed, as described in A, and is represented below the gel image (n = 2, representative image shown). (**I**) Steady-state kinetic analysis of TIR wild-type, oligomerization mutants (TIR^G747P and TIR^H833A), and catalytic mutants (TIR^E788Q and TIR^E788A) in 25% PEG 3350 with 0–2000 μM ε-NAD was assessed as described in D. The inset image outlined in red is an enlarged image of the mutant kinetic data. Kinetic parameters ($K_m$, $k_{cat}$, and $k_{cat}/K_m$) are shown in the table below the graph (n = 3). (**J, K**) SDS-PAGE analysis of TIR wild-type, oligomerization mutant (TIR^G747P) and catalytic mutants (TIR^E788Q and TIR^E788A) precipitation in the presence of 25% PEG 3350. Gel represents the soluble (**S**) and pellet (**P**) protein fractions of wild-type and mutant TIR following incubation with PEG 3350 and centrifugation. TIR^G747P and TIR^E788Q were assessed with 10 μM protein in J, and TIR^E788A was assessed with 3 μM protein in K (a lower concentration was used for TIR^E788A assays because the yield of the purified TIR^E788A mutant was low). Quantification of replicates represented below gel images (n = 4, representative images shown). *equals p < 0.05 by one-way ANOVA in J and unpaired t-test in K. (**L**) Effect of 1,6-hexanediol on TIR NADase activity is shown. TIR protein was incubated in the presence or absence of either 25% PEG 3350 or 500 mM citrate and treated with either 0 or 2% 1,6-hexanediol. The NADase activity of TIR for each condition was assessed using the ε-NAD substrate assay (n = 3). *equals p < 0.05 (unpaired t-test). (**M**) The NADase activity of TIR protein incubated with either 25% PEG 3350 or 500 mM citrate before (precentrifugation, n = 2) and after centrifugation, the supernatant (n = 4) and precipitant (n = 2) fractions. Precipitation fractions were resuspended in buffer alone or buffer containing 25% PEG 3350 or 500 mM citrate, and NADase activity was assessed. (**N**) Negative stain electron microscopy in either the absence or presence of 500 mM citrate (diameter of particles = 8.9 nm ± 1.2, n = 65). Representative circular particles are labeled with arrowheads. All error bars reflect SEM. See also *Figure 2—figure supplements 1 and 2*.

The online version of this article includes the following source data and figure supplement(s) for figure 2:

**Source data 1.** *Figure 2A* NADase activity of purified TIR at increasing TIR protein concentrations.

**Source data 2.** *Figure 2B* NADase activity of 2.5 μM TIR incubated in the presence of 25% (w/v) of macro- (PEG 8000, PEG 3350, and dextran) and micro- (sucrose and glycerol) viscogens.

**Source data 3.** *Figure 2C* Dose dependency of macroviscogens on the NADase activity of TIR.

**Source data 4.** *Figure 2D* Steady-state kinetic analysis of 2.5 μM TIR incubated in 0%–30% (w/v) of PEG 3350.

**Source data 5.** *Figure 2H* NADase activity of TIR protein in each fraction and at each concentration of PEG 3350.

**Source data 6.** *Figure 2I* Steady-state kinetic analysis of TIR wild-type, oligomerization mutant (TIR^G747P, TIR^H833A), and catalytic mutants (TIR^E788Q and TIR^E788A) in 25% PEG 3350.

**Source data 7.** *Figure 2J* Quantification of SDS-PAGE analysis of TIR: wild-type, oligomerization mutant (TIR^G747P) and catalytic mutant (TIR^E788Q), precipitation in the presence of 25% PEG 3350.

**Source data 8.** *Figure 2K* Quantification of SDS-PAGE analysis of TIR: wild-type and catalytic mutant (TIR^E788A), precipitation in the presence of 25% PEG 3350.

**Source data 9.** *Figure 2L* Effect of 1,6-hexanediol on TIR NADase activity.

**Source data 10.** *Figure 2M* NADase activity of TIR protein incubated with either 25% PEG 3350 or 500 mM citrate before (precentrifugation) and after centrifugation, the supernatant and precipitant fractions.

**Figure supplement 1.** TIR multimerization and phase transition superactivates its intrinsic NAD $^+$ glycohydrolase activity.

**Figure supplement 1—source data 1.** 1A enzyme concentration dependence in presence and absence of either 25% PEG 3350 or 500 mM sodium citrate.

**Figure supplement 1—source data 2.** 1B Dose dependency of citrate on the NADase activity of TIR.

**Figure supplement 1—source data 3.** 1C Steady-state kinetic analysis of 2.5 μM TIR incubated in 0–1000 mM sodium citrate.

**Figure supplement 1—source data 4.** 1G NADase activity of TIR protein in each fraction and at each concentration of citrate.

**Figure supplement 1—source data 5.** 1H Steady-state kinetic analysis of TIR wild-type, oligomerization mutant (TIR^G747P, TIR^H833A), and catalytic mutants (TIR^E788Q and TIR^E788A) in 500 mM sodium citrate.

**Figure supplement 1—source data 6.** 1I Quantification of SDS-PAGE analysis of TIR: wild-type, oligomerization mutant (TIR^G747P) and catalytic mutant (TIR^E788Q), precipitation in the presence of 500 mM sodium citrate.

**Figure supplement 1—source data 7.** 1J Quantification of SDS-PAGE analysis of TIR: wild-type and catalytic mutant (TIR^E788A), precipitation in the presence of 500 mM sodium citrate.

**Figure supplement 2.** TIR multimerization and phase transition superactivates its intrinsic NAD $^+$ glycohydrolase activity.

at which enzymes encounter their substrate (**Blacklow et al., 1988**). Interestingly, macroviscogens (polyethylene glycol [PEG] 3350 and PEG 8000), but not microviscogens (sucrose or glycerol), dramatically increased the NADase activity of TIR (**Figure 2B**). These effects were most pronounced with higher molecular weight PEGs, as treatment with smaller molecular weight PEGs (e.g. PEG 1500 and PEG 400) did not increase the enzymatic activity of TIR (**Figure 2C**). Specifically, PEGs 3350 and 8000 increase TIR activity in a concentration-dependent manner (**Figure 2C**). Crowding agents also increase the activity of the TIR domain of human SARM1, as well as plant TIR domains (**Horsefield et al., 2019**; **Loring et al., 2021**), suggesting that the mechanism of TIR regulation is strongly conserved.

Treatment with 25% PEG 3350 led to a roughly linear association between TIR concentration and NADase activity and markedly enhanced TIR activity at each enzyme concentration (**Figure 2—figure supplement 1A**). Consequently, lower amounts of enzyme (2.5 µM TIR) can be used to obtain robust kinetic data. Using these conditions, we determined the steady-state kinetic parameters ($k_{cat}$, $K_M$, and $k_{cat}/K_M$) with 2.5 µM TIR in the presence of increasing concentrations of PEG 3350 (**Figure 2D**). The $K_m$ of TIR increases and then decreases to level off at ~500 µM of ε-NAD with increasing concentration of PEG 3350 (**Figure 2E**). On the other hand, $k_{cat}$ increases nearly linearly with increasing concentrations of PEG 3350 (**Figure 2F**). In addition, the catalytic efficiency ($k_{cat}/K_m$) followed the same trend as $k_{cat}$, indicating that increased TIR activity is due to increased substrate turnover by the enzyme and not tighter substrate binding (**Figure 2G**).

We found that the NAD hydrolase activity of TIR is also activated by the precipitant sodium citrate, providing an orthologous method to characterize the enzymatic activity of the TIR domain (**Figure 2—figure supplement 1A and B**). Notably, the response to increasing citrate concentration was switch-like, where a concentration of at least 250 mM sodium citrate was needed to observe enzyme activity (**Figure 2—figure supplement 1B and C**). By contrast, activation with PEG 3350 was more dose-dependent (**Figure 2C and D**). Nevertheless, the kinetic parameters obtained in the presence of citrate displayed similar trends to those found with PEG 3350: $K_m$ decreased to level off at ~400 µM of ε-NAD (**Figure 2—figure supplement 1D**), $k_{cat}$ increases with increasing sodium citrate concentration (**Figure 2—figure supplement 1E**), and $k_{cat}/K_m$ follows the $k_{cat}$ trends (**Figure 2—figure supplement 1F**).

In the cytoplasm, high concentrations of macromolecules (proteins, nucleic acids, lipids, carbohydrates) cause molecular crowding and induce phase transitions of signaling proteins (**Rivas and Minton, 2016**). To determine if TIR undergoes a phase transition and whether its NADase activity correlates with the transition, we incubated purified TIR with different concentrations of PEG3350 or citrate, centrifuged the sample, and evaluated TIR NADase activity in the soluble and insoluble fractions (**Figure 2H** and **Figure 2—figure supplement 1G**). At low concentrations of both PEG and citrate, TIR protein was present in the supernatant. However, at high concentrations of both PEG and citrate, TIR was principally located in the pelleted (insoluble) fraction, as visualized by Coomassie staining on an SDS-PAGE gel (**Figure 2H** and **Figure 2—figure supplement 1G**). Importantly, robust NADase activity was observed in the pelleted fraction and not in the supernatant of samples treated with high concentrations of PEG or citrate (**Figure 2H** and **Figure 2—figure supplement 1G**). Taken together, these data demonstrate that precipitation/aggregation of TIR is required to activate the intrinsic NADase activity of TIR-1/SARM1.

To determine whether TIR-1 multimerization is also required for a phase transition and the NADase activity of TIR, we recombinantly expressed and purified the TIR domain of TIR-1, which contained mutations in residues required for oligomerization (TIR^G747P and TIR^H833A) and NAD⁺ hydrolysis (TIR^E788A and TIR^E788Q). Notably, TIR^G747P and TIR^E788Q mutants showed no apparent activity, and TIR^E788A and TIR^H833A showed a > 1 × 10⁸fold decrease in catalytic efficiency ($k_{cat}/K_m$) compared to TIR^WT (**Figure 2I**). Similarly, kinetic analysis of TIR oligomerization and catalytic mutants in 500 mM citrate revealed that all the mutants were catalytically dead, each showing no apparent activity (**Figure 2—figure supplement 1H**).

To further characterize the TIR phase transition, we evaluated the precipitation capacity of TIR oligomerization and catalytic mutants in the presence of PEG 3350 and citrate. Consistent with our genetic data, TIR^G747P, which contains a mutation that prevents oligomerization of TIR, was unable to precipitate as readily as TIR^WT (**Figure 2J**). Compared to TIR^WT, the TIR^G747P mutant in the presence of PEG 3350 demonstrated a 25% decrease in the amount of precipitated protein (**Figure 2J**). We observed similar results in the presence of citrate – compared to TIR^WT, the TIR^G747P mutant demonstrated a 22%

decrease in precipitated protein compared to TIR^WT (*Figure 2—figure supplement 1I*). Importantly, TIR proteins with two different mutations in the glutamate required for NAD$^+$ catalysis, TIR^E788A and TIR^E788Q, precipitated to a similar extent as TIR^WT (*Figure 2J and K*, and *Figure 2—figure supplement 1I and J*) but had minimal NADase activity (*Figure 2I* and *Figure 2—figure supplement 1H*). Collectively, these data demonstrate that a phase transition of TIR-1 is required for its NADase activity.

During phase separations, macromolecules partition into distinct biochemical compartments characterized by a higher concentration dense phase and a lower concentration dilute phase (*Boeynaems et al., 2018*; *Alberti and Dormann, 2019*; *Verdile et al., 2019*). The dense compartment can have either liquid-like properties in liquid-to-liquid phase separations or solid-like properties in liquid-to-solid phase transitions (*Alberti and Dormann, 2019*; *Patel et al., 2015*; *Kroschwald et al., 2015*; *Peskett et al., 2018*). To determine whether TIR undergoes a liquid-to-liquid or a liquid-to-solid phase transition, we assayed the enzyme activity of TIR in the presence of 1,6-hexanediol, an aliphatic alcohol that interferes with hydrophobic interactions prominent in liquid-to-liquid separations. Thus, 1,6-hexanediol disrupts liquid-like compartments, but not solid-like compartments (*Boeynaems et al., 2018*; *Verdile et al., 2019*; *Kroschwald et al., 2015*). In the presence of either PEG 3350 or citrate, we observed that 1,6-hexanediol decreases the NADase activity of TIR by <2 fold (*Figure 2L*). While significant, this modest decrease (the activity of 1,6-hexanediol-treated TIR remains two orders of magnitude higher than the activity of TIR without PEG 3350 or citrate addition) suggests that the NADase activity of TIR is predominantly associated with a solid, rather than a liquid, state or lies on a continuum between a liquid-like and solid-like state (*Figure 2L*). Notably, in the absence of PEG 3350 or citrate, the low-level TIR NADase activity can be inhibited by 1,6-hexanediol (*Figure 2L*), suggesting that this activity is driven by hydrophobic interactions. Importantly, 1,6-hexanediol does not interfere with TIR precipitation in the presence of PEG or citrate, regardless of whether it is added before or after precipitate formation (*Figure 2—figure supplement 2A and B*).

Next, we determined if this liquid-to-solid phase transition of TIR is reversible. The insoluble fraction following treatment of TIR with PEG 3350 or citrate was resuspended in either buffer alone or buffer plus the respective additive (PEG 3350 or citrate). If the phase transition is reversible, resuspension of precipitated TIR in buffer alone should disrupt enzymatic activity. However, if the phase transition is irreversible, activity should be detected when the pellet is resuspended in buffer alone. Notably, we observed some TIR enzymatic activity when the pellet was resuspended in buffer (*Figure 2M*). With PEG 3350, the activity was lower than that observed in the precentrifugation control or when the pellet was resuspended in buffer with PEG 3350. By contrast, with citrate, the activity was similar to both control conditions (precentrifugation or when the pellet was resuspended in buffer and citrate). These data indicate that the phase transition of TIR is partially reversible in PEG 3350 and irreversible in sodium citrate. To confirm these findings, we centrifuged the resuspended samples to examine the fractions visually by SDS-PAGE (*Figure 2—figure supplement 2C*). In the sample initially prepared with PEG 3350 and resuspended in buffer alone, a faint band was present in the supernatant fraction. However, this band was absent in the sample initially prepared with sodium citrate (*Figure 2—figure supplement 2C*). These data confirm that the TIR phase transition is partially reversible in PEG 3350 and irreversible in sodium citrate, at least under these conditions.

Next, we evaluated the effect of pH on TIR precipitation and NADase activity. There was virtually no increase in TIR precipitation in the absence of PEG. However, in the presence of 25% PEG, TIR precipitation increased with increasing pH (*Figure 2—figure supplement 2D*). Under these same conditions, no activity was apparent below pH 6, $K_m$ remained constant (*Figure 2—figure supplement 2E*), and $k_{cat}$ increased from pH 6.5 to pH 8.5 (*Figure 2—figure supplement 2F*). A corresponding increase in $k_{cat}/K_m$ was responsible for the increase in catalytic efficiency above pH 7 (*Figure 2—figure supplement 2G*). Notably, the increase in $k_{cat}/K_m$ correlated with precipitation (*Figure 2—figure supplement 2D and G*), again indicating that a phase transition increases TIR activity.

We performed negative stain electron microscopy to directly visualize TIR aggregation in vitro (*Figure 2N*). Protein visualization is not possible with PEG because macroviscogens themselves are stained, confounding image analysis. Therefore, we performed this experiment with citrate. In the absence of citrate, we observed borderline fibrillar structures and protein aggregates, but overall, there were no consistent structures (*Figure 2N*). However, in the presence of citrate, circular particles emerged (*Figure 2N*). These data corroborate our discovery that TIR-1::wrmScarlet aggregates in vivo into visible puncta within intestinal epithelial cells (*Figure 1A*).

## Cholesterol deficiency activates the *C. elegans* p38 immune pathway through the multimerization and NADase activity of TIR-1/SARM1

*C. elegans* is a sterol auxotroph and requires dietary sterols for development, lifespan, fecundity, and resistance to pathogen infection (*Hieb and Rothstein, 1968*; *Chitwood, 1999*; *Shim et al., 2002*; *Merris et al., 2003*; *Yochem et al., 1999*; *Otarigho and Aballay, 2020*; *Cheong et al., 2011*). As such, 5 µg/mL of cholesterol is a standard additive in *C. elegans* laboratory growth medium (*Brenner, 1974*). We found that *C. elegans* grown in the absence of cholesterol supplementation activated GFP-based transcriptional reporters for two putative immune effector genes, T24B8.5p::*gfp* and *irg-5*p::*gfp* (*Figure 3A and B*). T24B8.5 and *irg-5* are expressed in the intestine, induced during infection with multiple pathogens, including *P. aeruginosa,* and controlled by the p38 PMK-1 innate immune pathway (*Troemel et al., 2006*; *Shivers et al., 2009*; *Bolz et al., 2010*). qRT-PCR studies confirmed that *C. elegans* in a low cholesterol environment upregulate T24B8.5 and *irg-5,* as well as other innate immune effector genes (*irg-4* and K08D8.4) (*Figure 3C*). These data suggest that host defense pathways are activated in the absence of pathogen infection when environmental sterols are scarce.

The nuclear hormone receptor, NHR-8, a homolog of mammalian liver X receptor (LXR) and pregnane X receptor (PXR), is required for the transport, distribution, and metabolism of cholesterol in *C. elegans* (*Magner et al., 2013*; *Lindblom et al., 2001*). Thus, *nhr-8* loss-of-function mutant strains can be used as genetic tools to study conditions of low sterol content. Two previously characterized *nhr-8* null alleles are *nhr-8(hd117)*, which lacks the first exon (*Magner et al., 2013*) and *nhr-8(ok186)*, which is missing most of the ligand-binding domain (*Lindblom et al., 2001*). Notably, the transcription profile of *nhr-8(hd117)* and *nhr-8(ok186)* animals mimics that of wild-type *C. elegans* infected with the bacterial pathogen *P. aeruginosa* (*Figure 3D*, *Figure 3—figure supplement 1A*, *Supplementary files 1 and 2*). The correlation between the transcriptional signatures of either the *nhr-8(hd117)* or the *nhr-8(ok186)* mutant with the genes that are changed in wild-type animals during pathogen infection was significant across all genes ($r = 0.311$ and $r = 0.370$, respectively). Of note, the correlation between these datasets is tighter when comparing only the differentially expressed genes ($r = 0.763$ and $r = 0.849$, respectively) and only genes that are also involved in innate immunity ($r = 0.677$ and $r = 0.703$, respectively) (*Figure 3D* and *Figure 3—figure supplement 1A*). Among the immune effectors that are upregulated in both the *nhr-8(hd117)* and *nhr-8(ok186)* mutants, and in wild type animals infected with *P. aeruginosa,* are T24B8.5, *irg-4*, *irg-5*, and K08D8.4; the same genes whose transcription are also induced by cholesterol deprivation (*Figure 3C and D*, and *Figure 3—figure supplement 1A*).

To validate these findings, we compared gene expression changes in wild-type *C. elegans* grown in media lacking supplemented cholesterol versus animals grown under standard culture conditions with 5 µg/mL of cholesterol. Consistent with our transcriptome profiling experiments of *nhr-8* loss-of-function mutants, we discovered that innate immune effectors were strongly enriched among genes that were transcriptionally activated during cholesterol deprivation (*Figure 3E*). Specifically, the expression of innate immune effectors was significantly correlated in wild-type animals infected with *P. aeruginosa* and in wild-type nematodes starved for cholesterol ($r = 0.610$) (*Figure 3E*). Immune effectors that were induced during cholesterol starvation in qRT-PCR studies (*irg-4*, *irg-5*, K08D8.4, and T24B8.5) were again found among these differentially regulated genes, confirming the integrity of our RNA-seq analysis (*Figure 3C*). These observations were confirmed in a separate RNA-seq experiment (*Otarigho and Aballay, 2021*). Importantly, a comparison of the transcriptional changes in *nhr-8(hd117)* mutants with the genes regulated by low cholesterol revealed a remarkable correlation across all genes ($r = 0.578$) and differentially expressed genes ($r = 0.836$) (*Figure 3—figure supplement 1B*). Together with previous work characterizing the role of *nhr-8* in *C. elegans* cholesterol metabolism (*Magner et al., 2013*), these data support the use of *nhr-8* mutants to study conditions of low sterol content. In summary, cholesterol-starved, wild-type *C. elegans* and two different *nhr-8* loss-of-function mutants induce the transcription of innate immune defenses.

Two different supplementation experiments using the T24B8.5p::*gfp* immune reporter demonstrated that immune effector activation in *nhr-8(hd117)* mutants is due to sterol deficiency in these animals. First, supplementation of exogenous cholesterol at an increased concentration (80 µg/mL) fully suppressed T24B8.5p::*gfp* activation in the *nhr-8(hd117)* mutant (*Figure 3F*). Second, supplementation with the non-ionic detergents Tergitol or Triton X-100, which solubilize hydrophobic, amphipathic compounds, including sterols, suppressed T24B8.5p::*gfp* activation in *nhr-8(hd117)* animals in a manner that was dependent on the presence of added cholesterol in the growth media (*Figure 3G*



**Figure 3.** Cholesterol scarcity activates intestinal innate immune defenses. Images of T24B8.5p::*gfp* (**A**) and *irg-5*p::*GFP* (**B**) transcriptional immune reporters in wild-type animals grown on standard nematode growth media ( + 5 µg/mL cholesterol) and in the absence of supplemented cholesterol ( + 0 µg/mL cholesterol). (**C**) qRT-PCR data of the indicated innate immune effector genes in wild-type *C. elegans* grown in the presence ( + 5 µg/mL) and absence ( + 0 µg/mL) of supplemented cholesterol. *equals p < 0.05 (unpaired t-test). (**D and E**) Data from mRNA-seq experiments comparing genes

*Figure 3 continued on next page*

*Figure 3 continued*

differentially regulated in uninfected *nhr-8(hd117)* mutants versus wild-type animals (**D**) or uninfected wild-type animals grown in the absence (0 µg/mL) versus presence (5 µg/mL) of supplemental cholesterol (**E**) (y-axis) are compared with genes differentially expressed in wild-type animals during *P. aeruginosa* infection (x-axis). All genes are shown in gray. Genes that are differentially expressed in both datasets are shown in black (Fold change >2, q < 0.01). Genes that are annotated as innate immune genes are shown in red. The location of the representative genes T24B8.5, *irg*-5, *irg*-4, and K08D8.4, whose expression is examined throughout this manuscript, are shown. (Of note, in the 0 µg/mL cholesterol mRNA-seq data set K08D8.4 did not meet our cut-off threshold, although was significantly upregulated, fold change = 1.79, q = 5.6 × 10⁻⁴). See also *Supplementary files 1-3*. (**F, G, H**) Images of T24B8.5p::*gfp* animals of the indicated genotypes grown under the indicated conditions are shown. (**H**) *C. elegans* were grown on media solidified with agarose rather than agar. (**I, J, K, L**) qRT-PCR data of the indicated genes in wild-type and *nhr-8(hd117)* mutant animals grown on standard nematode growth media ( + 5 µg/mL cholesterol) in the presence or absence of 0.1% Tergitol, as indicated. For the qRT-PCR studies in (**C, I, J, K and L**), data are the average of three to six independent biological replicates, each normalized to a control gene with error bars representing SEM and are presented as the value relative to the average expression from all replicates of the indicated gene in wild-type animals on standard nematode growth media ( + 5 µg/mL cholesterol). *equals p < 0.05 (two-way ANOVA with Tukey's multiple comparison testing). (**M, N**) Survival curves for *C. elegans* pathogenesis assays with *P. aeruginosa* and *C. elegans* of the indicated genotypes at the L4 larval stage and exposed to the indicated conditions. Data are representative of three trials. The difference between the *nhr-8(hd117)* mutant and the other conditions in both M and N is significant (p < 0.05). The Kaplan-Meier method was used to estimate the survival curves for each group, and the log-rank test was used for all statistical comparisons. Sample sizes, mean lifespan and p-values for all trials are shown in *Supplementary file 4*. (**O**) *P. aeruginosa,* isolated from the intestines of animals with the indicated genotypes, were quantified after 24 hr of bacterial infection. Data are colony-forming units (CFU) of *P. aeruginosa* and are presented as the average of 10 separate biological replicates, with each replicate containing 10–11 animals. *equals p < 0.05 (unpaired t-test). Scale bars in all images equal 200 µm. See also *Figure 3—figure supplement 1*.

The online version of this article includes the following source data and figure supplement(s) for figure 3:

**Source data 1.** *Figure 3C* qRT-PCR data of the indicated innate immune effector genes in wild-type *C. elegans* growing in the presence (+5 µg/mL) and absence (+0 µg/mL) of supplemented cholesterol.

**Source data 2.** *Figure 3I* qRT-PCR data of T24B8.5 in wild-type and *nhr-8(hd117)* mutant animals grown on standard nematode growth media (+5 µg/mL cholesterol) in the presence or absence of 0.1% Tergitol, as indicated.

**Source data 3.** *Figure 3J* qRT-PCR data of *irg-4* in wild-type and *nhr-8(hd117)* mutant animals grown on standard nematode growth media (+5 µg/mL cholesterol) in the presence or absence of 0.1% Tergitol, as indicated.

**Source data 4.** *Figure 3K* qRT-PCR data of *irg-5* in wild-type and *nhr-8(hd117)* mutant animals grown on standard nematode growth media (+5 µg/mL cholesterol) in the presence or absence of 0.1% Tergitol, as indicated.

**Source data 5.** *Figure 3L* qRT-PCR data of K08D8.4 in wild-type and *nhr-8(hd117)* mutant animals grown on standard nematode growth media (+5 µg/mL cholesterol) in the presence or absence of 0.1% Tergitol, as indicated.

**Source data 6.** *Figure 3O* Colony-forming units (CFUs) of *P. aeruginosa*, isolated from the intestines of animals with the indicated genotypes, quantified after 24 hr of bacterial infection.

**Figure supplement 1.** Cholesterol scarcity activates intestinal innate immune defenses.

*and H*, and *Figure 3—figure supplement 1C*). Nematode growth media solidified with agarose, rather than agar, contains markedly fewer contaminating sterols (*Magner et al., 2013*; *Chitwood, 1999*; *Gerisch et al., 2001*). Importantly, on agarose, the addition of Tergitol to the growth media also suppressed T24B8.5p::*gfp* activation in the *nhr-8(hd117)* mutant, but only in the presence of 5 µg/mL of cholesterol (*Figure 3H*). Notably, this effect was dose-dependent (*Figure 3—figure supplement 1C*). These data establish that solubilization of cholesterol by Tergitol in standard nematode growth media suppresses immune activation in the *nhr-8(hd117)* mutant background. We used qRT-PCR to confirm these findings – Tergitol suppressed activation of T24B8.5, as well as the immune effectors *irg-4*, *irg-5*, and K08D8.4 in the *nhr-8(hd117)* mutant (*Figure 3I–L*). Of note, activation of immune defenses in *nhr-8(hd117)* animals is specific to cholesterol deprivation in this genetic background, as supplementation with individual saturated, mono- and polyunsaturated fatty acids, which are also misregulated in the *nhr-8(hd117)* mutant background (*Magner et al., 2013*), failed to suppress activation of T24B8.5p::*gfp* (*Figure 3—figure supplement 1D*). Together, these data confirm that cholesterol deprivation activates immune effector transcription independent of bacterial infection.

It has been previously shown that wild-type *C. elegans* raised on media without supplemented cholesterol are hypersusceptible to killing by *P. aeruginosa*, as are *nhr-8(hd117)* mutants on media with standard cholesterol supplementation (5 µg/mL) (*Otarigho and Aballay, 2020*). Importantly, we found that high doses of cholesterol (80 µg/mL) rescued the enhanced susceptibility of *nhr-8(hd117)* mutants to *P. aeruginosa* infection (*Figure 3M*). We confirmed this finding using an orthologous approach. Two detergents (Tergitol or Triton X-100), which solubilize cholesterol in nematode growth media, also

restored wild-type pathogen resistance to the *nhr-8(hd117)* mutant (*Figure 3N*), consistent with their ability to modulate the expression of innate immune effectors (*Figure 3G–L*). Importantly, we found that the *nhr-8(hd117)* mutant accumulated less *P. aeruginosa* in its intestine during infection than wild-type animals (*Figure 3O*). Taken together, these data suggest that low cholesterol in the *nhr-8(hd117)* mutants reduces tolerance to pathogen infection, and that the robust transcriptional induction of immune effectors in this mutant background leads to reduced accumulation of *P. aeruginosa* during infection.

Additional analyses of the transcriptome profiling data revealed that targets of the p38 PMK-1 innate immune pathway were strongly enriched among the genes induced in wild-type animals during cholesterol deprivation (*Figure 4A*). Accordingly, the levels of active, phosphorylated PMK-1 were higher in wild-type animals grown on media without supplemented cholesterol than in animals grown under standard culture conditions (*Figure 4B and C*). Consistent with these data, the p38 PMK-1 pathway is also activated in *nhr-8* loss-of-function mutants. A gene set enrichment analysis of the *nhr-8(hd117)* and *nhr-8(ok186)* transcriptome profiling experiments revealed strong enrichment of p38 PMK-1 pathway targets among the genes upregulated in each *nhr-8* mutant (*Figure 4D* and *Figure 4— figure supplement 1A*). In addition, the *nhr-8(hd117)* and *nhr-8(ok186)* mutants had an increased ratio of phosphorylated PMK-1 relative to total PMK-1 compared to wild-type controls (*Figure 4E and F*). RNAi-mediated knockdown of *tir-1*, the most upstream component of the p38 signaling cassette (*Liberati et al., 2004*; *Couillault et al., 2004*), fully suppressed hyperactivation of T24B8.5p::*gfp* in *nhr-8(hd117)* animals (*Figure 4G*). The *tir-1(qd4)* loss-of-function mutation completely suppressed the induction of T24B8.5, *irg-4*, *irg-5*, and K08D8.4 in the *nhr-8(hd117)* background (*Figure 4H–K*). The transcription factors ATF-7 and SKN-1 link PMK-1 activity to its transcriptional outputs (*Shivers et al., 2010*; *Inoue et al., 2005*). In *nhr-8(hd117)* animals, knockdown of *atf-7*, but not *skn-1*, abrogated T24B8.5p::*gfp* activation (*Figure 4G* and *Figure 4—figure supplement 1B and C*). Finally, to further support our observation that cholesterol scarcity induces immune defenses upstream of p38 PMK-1, we used the MAPK phosphatase *vhp-1*, a negative regulator of PMK-1 (*Kim et al., 2004*). Solubilization of cholesterol with Tergitol was unable to suppress activation of T24B8.5p::*gfp* induced by knockdown of *vhp-1* (*Figure 4L*).

The c-JUN N-terminal kinase MAPK homolog *kgb-1*, the insulin signaling pathway forkhead box O family (FOXO) transcription factor *daf-16*, and the G-protein-coupled receptor (GPCR) *fshr-1*, each function in parallel to the p38 PMK-1 pathway to regulate immune and stress responses in *C. elegans* (*Troemel et al., 2006*; *Kim et al., 2004*; *Powell et al., 2009*; *Garsin et al., 2003*). However, knockdown of each of these genes in *nhr-8(hd117)* animals failed to suppress T24B8.5p::*gfp* activation (*Figure 4—figure supplement 1B*), and RNAi-mediated knockdown of *nhr-8* did not induce nuclear localization of DAF-16::GFP (*Figure 4—figure supplement 1D*). Thus, cholesterol scarcity induces *C. elegans* innate immune responses through specific activation of p38 PMK-1 immune pathway signaling.

Interestingly, *tir-1(qd4);nhr-8(hd117)* double mutants were more susceptible to *P. aeruginosa* infection than *nhr-8(hd117)* mutants (*Figure 4M*). These data suggest that the induction of the p38 PMK-1 pathway in the *nhr-8(hd117)* mutant background promotes resistance to *P. aeruginosa* infection. In addition, *tir-1(qd4);nhr-8(hd117)* double mutants were slightly, but significantly and reproducibly, more susceptible than the *tir-1(qd4)* mutant to killing by *P. aeruginosa*. We therefore hypothesize that the inherent susceptibility to pathogen-mediated killing in animals that lack sufficient cholesterol (e.g. the *nhr-8(hd117)* mutant) leads to additive pathogen susceptibility in animals that also lack a functioning p38 PMK-1 host defense pathway (e.g. the *tir-1(qd4)* mutant).

To determine if sterol scarcity induces oligomerization and the NAD$^+$ glycohydrolase activity of TIR-1 to activate the p38 PMK-1 pathway, we examined TIR-1::wrmScarlet puncta formation in the *nhr-8(hd117)* mutant background (*Figure 5A*). Consistent with our studies during *P. aeruginosa* infection (*Figure 1A*), *nhr-8(hd117)* mutant animals induced TIR-1::wrmScarlet puncta formation in intestinal epithelial cells (*Figure 5A and B*). Importantly, TIR-1::wrmScarlet puncta formation in *nhr-8(hd117)* animals was entirely suppressed by solubilizing cholesterol in the growth media with Tergitol (*Figure 5A and B*). Additionally, treatment of *nhr-8(hd117)* animals expressing TIR-1::wrmScarlet with *tir-1(RNAi)* inhibited fluorescence and puncta formation (*Figure 5A and B*).

Importantly, the *C. elegans tir-1*$^{ΔSAM}$, *tir-1*$^{G747P}$ and *tir-1*$^{H833A}$ mutants, which contain specific mutations that block oligomerization of TIR-1, and the *tir-1*$^{E788A}$ strain that carries a mutation in the catalytic



**Figure 4.** Cholesterol scarcity activates the p38 PMK-1 innate immune pathway. (**A**) Gene set enrichment analysis (GSEA) of p38 PMK-1 targets in the 0 μg/mL cholesterol mRNA-seq experiment. Fold change in expression of significantly differentially expressed genes (fold-change > 2 and q < 0.01) in uninfected animals grown in the absence (0 μg/mL) versus presence (5 μg/mL) of supplemental cholesterol are ranked from higher expression (red) to lower expression (blue). Normalized enrichment score (NES) and q-value are indicated. p38 PMK-1 targets found in the transcriptional profile are indicated by hit number in the left margin and black lines. (**B**) An immunoblot analysis of lysates from wild-type *C. elegans* grown on standard nematode

*Figure 4 continued on next page*

*Figure 4 continued*

growth media in the presence ( + 5 µg/mL cholesterol) and in the absence ( + 0 µg/mL cholesterol) of supplemented cholesterol using antibodies that recognize the doubly phosphorylated TGY motif of PMK-1 (phos-PMK-1), total PMK-1 protein (total PMK-1), and tubulin (α-tubulin) is shown. PMK-1 is a 43.9 kDa protein and tubulin is a 50 kDa protein. (**C**) The band intensities of four biological replicates of the Western blot shown in A were quantified. Error bars represent SEM. *equals p < 0.05 (unpaired t-test) (**D**) GSEA of p38 PMK-1 targets in the *nhr-8* mRNA-seq experiment as described in A. (**E, F**) Western blot experiment (**E**) and quantification (**F**) of four biological replicate experiments as described in B and C with the strains of the indicated genotypes. In B and E, *pmk-1(km25)* and *nsy-1(ag3)* are loss-of-function mutants that serve as controls to confirm the specificity of phos-PMK-1 and total PMK-1 probing. Error bars represent SEM. *equals p < 0.05 (one-way ANOVA with Dunnett's multiple comparison testing). (**G**) Images of T24B8.5p::*gfp* transcriptional immune reporter expression in wild-type animals and in *nhr-8(hd117)* mutants grown on control RNAi, *tir-1(RNAi)* or *atf-7(RNAi)* bacteria, as indicated. Quantification of GFP expression for this experiment is presented in *Figure 4—figure supplement 1C*. (**H, I, J, K**) qRT-PCR data of the indicated genes in the indicated mutant animals grown on standard nematode growth media ( + 5 µg/mL cholesterol). Data are the average of three to seven independent replicates, each normalized to a control gene with error bars representing SEM and are presented as the value relative to the average expression from all replicates of the indicated gene in wild-type animals. *equals p < 0.05 (one-way ANOVA with Dunnett's multiple comparison testing) (**L**) Images of T24B8.5p::*gfp* transcriptional immune reporter animals of the indicated genotypes grown in the presence or absence of 0.1% Tergitol. Scale bars in all images equal 200 µm. (**M**) Survival curves of *C. elegans* pathogenesis assay with *P. aeruginosa* and *C. elegans* of the indicated genotypes at the L4 larval stage exposed to the indicated conditions. Data are representative of three trials. Difference between *nhr-8(hd117)* and all other conditions is significant (p < 0.05). The Kaplan-Meier method was used to estimate the survival curves for each group, and the log-rank test was used for all statistical comparisons. Sample sizes, mean lifespan, and p-values for all trials are shown in *Supplementary file 4*. See also *Figure 4—figure supplement 1*.

The online version of this article includes the following source data and figure supplement(s) for figure 4:

**Source data 1.** *Figure 4C* Quantification of p38 immunoblot analysis of lysates from wild-type *C. elegans* grown on standard nematode growth media in the presence ( + 5 µg/mL cholesterol) and in the absence ( + 0 µg/mL cholesterol) of supplemented cholesterol.

**Source data 2.** *Figure 4F* Quantification of p38 immunoblot analysis of lysates from the indicated *C. elegans* strains.

**Source data 3.** *Figure 4H* qRT-PCR data of T24B8.5 in the indicated mutant animals grown on standard nematode growth media ( + 5 µg/mL cholesterol).

**Source data 4.** *Figure 4I* qRT-PCR data of *irg-4* in the indicated mutant animals grown on standard nematode growth media ( + 5 µg/mL cholesterol).

**Source data 5.** *Figure 4J* qRT-PCR data of *irg-5* in the indicated mutant animals grown on standard nematode growth media ( + 5 µg/mL cholesterol).

**Source data 6.** *Figure 4K* qRT-PCR data of K08D8.4 in the indicated mutant animals grown on standard nematode growth media ( + 5 µg/mL cholesterol).

**Figure supplement 1.** Cholesterol scarcity activates the p38 PMK-1 innate immune pathway.

**Figure supplement 1—source data 1.** 1C Quantification of T24B8.5p::*gfp* expression in the indicated conditions.

glutamine required for the NADase activity, each prevented activation of the p38 PMK-1-dependent immune reporter T24B8.5p::*gfp* in *nhr-8(RNAi)* animals (*Figure 5C*) and during cholesterol deprivation (*Figure 5D*). Using qRT-PCR, we confirmed that these mutations in *tir-1* abrogated the induction of the T24B8.5 (*Figure 5E*) and *irg-4* (*Figure 5F*) immune effectors in *nhr-8(RNAi)* animals. We also found that these *tir-1* mutants phenocopied the effects of the p38 PMK-1 pathway mutants (*tir-1(qd4)*, *nys-1(ag3)*, and *pmk-1(km25)*) on the basal expression of these immune effector genes (*Figure 5E and F*).

It is possible that the organization of TIR-1::wrmScarlet into visible puncta in *nhr-8* mutants is secondary to non-specific protein aggregation; however, the in vitro and in vivo data presented in this manuscript, when considered together, suggest that this is not the case. Organized multimerization of TIR-1 is a prerequisite for the NADase activity of the protein complex in vitro (*Figure 2*, *Figure 2—figure supplement 1*). Accordingly, the mutations that specifically abrogate TIR-1 multimerization in vitro (*Figure 2J*, *Figure 2—figure supplement 1I*), also block p38 PMK-1 pathway activation (*Figure 1D–F*) and prevented immune effector induction in response to cholesterol deprivation in vivo (*Figure 5C–F*). Furthermore, Tergitol, which solubilizes cholesterol and fully suppresses p38 PMK-1 immune activation in *nhr-8* mutants (*Figure 3G-N*), also suppresses TIR-1::wrmScarlet puncta formation (*Figure 5*). Finally and perhaps most importantly, *P. aeruginosa* infection, a separate physiological stress, also induces TIR-1::wrmScarlet puncta formation to activate the p38 PMK-1 pathway (*Figure 1*).

In summary, the above data demonstrate for the first time that physiological stresses, both pathogen and non-pathogen, induce TIR-1 multimerization into puncta within intestinal epithelial cells, which then activates the p38 PMK-1 innate immune pathway through the intrinsic NADase activity of the TIR-1 protein complex.



**Figure 5.** Sterol stress activates the *C. elegans* p38 immune pathway through the multimerization and NAD $^+$ glycohydrolase activity of TIR-1/SARM1. (**A**) Images of *tir-1::wrmScarlet* and *nhr-8(hd117);tir-1::wrmScarlet* animals as described in *Figure 1A* exposed to either 0.1% Tergitol or *tir-1(RNAi)*. Scale bar equals 20 μm (2 μm for the inset enlarged images). (**B**) Quantification of the number of puncta present in the red (TIR-1::wrmScarlet), but not green (autofluorescence) channel with indicated conditions as described in *Figure 1B*. *equals p < 0.05 (two-way ANOVA with Tukey's multiple

*Figure 5 continued on next page*

*Figure 5 continued*

comparison testing). (**C,D**) Images of T24B8.5p::*gfp* transcriptional immune reporter in *tir-1* mutants with defects in oligomerization (*tir-1*ΔSAM, *tir-1*G747P, and *tir-1*H833A) and NADase catalytic activity (*tir-1*E788A) following *nhr-8(RNAi)* (**C**), and during cholesterol deprivation (**D**). Scale bar equals 200 µm. (**E,F**) qRT-PCR data of T24B8.5 (**E**) and *irg-4* (**F**) in wild-type and mutant animals of the indicated genotypes grown on standard nematode growth media ( + 5 µg/mL cholesterol). Data are the average of three independent replicates, each normalized to a control gene with error bars representing SEM and are presented as the value relative to the average expression from all replicates in wild-type animals. *equals p < 0.05 (two-way ANOVA with Tukey's multiple comparison testing).

The online version of this article includes the following source data for figure 5:

**Source data 1.** *Figure 5B* Quantification of the number of TIR-1::wrmScarlet puncta present in the last posterior pair of intestinal epithelial cells in the indicated strains and conditions.

**Source data 2.** *Figure 5E* qRT-PCR data of T24B8.5 in wild-type and mutant animals of the indicated genotypes grown on standard nematode growth media ( + 5 µg/mL cholesterol).

**Source data 3.** *Figure 5F* qRT-PCR data of *irg-4* in wild-type and mutant animals of the indicated genotypes grown on standard nematode growth media ( + 5 µg/mL cholesterol).

## Sterol scarcity primes p38 PMK-1 immune defenses

Since *C. elegans* requires cholesterol to survive bacterial infection and must obtain this essential metabolite from its diet, we hypothesized that, when environmental sterols are limited, activation of the p38 PMK-1 pathway represents an evolutionary adaptation that primes immune effector expression to anticipate challenges from bacterial pathogens. To test this hypothesis, we examined the expression of innate immune effector genes during bacterial infection in the presence and absence of cholesterol supplementation. Interestingly, the induction of *irg-4*p::*gfp* (*Figure 6A*), *irg-5*p::*gfp* (*Figure 6B*), and T24B8.5p::*gfp* (*Figure 6C*) during *P. aeruginosa* infection was enhanced when nematodes were infected on media that did not contain supplemented cholesterol. Consistent with these data, *P. aeruginosa* infection also led to increased activation of *irg-4*p::*gfp* (*Figure 6D*) and *irg-5*p::gfp (*Figure 6E*) when *nhr-8* was depleted by RNAi.

To provide further support for this hypothesis, we analyzed the expression pattern of p38 PMK-1-dependent transcripts in *nhr-8(hd117)* mutant animals during *P. aeruginosa* infection. Of the 472 genes that were induced in wild-type animals during *P. aeruginosa* infection, 184 were also differentially regulated (either induced or repressed) in *nhr-8(hd117)* animals that were infected with *P. aeruginosa*. To perform this analysis in an unbiased manner, we scaled the expression level in each condition for each of these 184 genes by calculating a row z-score and performed hierarchical clustering (*Figure 6F*). We observed that these 184 genes group into two clusters: Cluster I had 139 genes and Cluster II contained 45 genes (*Supplementary file 5*). Cluster I was comprised mostly of genes whose expression in the absence of infection was higher in *nhr-8(hd117)* mutants than wild-type animals (129 of 139 genes). In addition, the majority of Cluster I genes were more strongly induced during *P. aeruginosa* infection in *nhr-8(hd117)* animals than wild-type animals. Cluster II, on the other hand, contained genes whose induction on *P. aeruginosa* were dependent on *nhr-8*. Importantly, p38 PMK-1-dependent genes were strongly enriched among Cluster I, but not Cluster II, genes (*Figure 6G*). Thirty-three of the 139 genes in Cluster I are known targets of the p38 PMK-1 immune pathway (24.17-fold enriched, hypergeometric *P*-value = 7.68 x 10⁻³⁷), a group that includes the immune effectors T24B8.5 (*Figure 6H*), *irg-4* (*Figure 6I*), *irg-5* (*Figure 6J*), and K08D8.4 (*Figure 6K*). By contrast, only three genes in Cluster II are p38 PMK-1-dependent transcripts (6.95-fold enriched, hypergeometric p-value = 0.01) (*Figure 6F and G*).

These data demonstrate that sterol scarcity primes the induction of innate immune effectors by activating the p38 PMK-1 immune pathway. Subsequent challenge by *P. aeruginosa* further drives immune activation in a manner that leads to reduced pathogen accumulation in the intestine (*Figure 3O*).

## Discussion

Here, we report two conceptual advances. First, we show that a phase transition of *C. elegans* TIR-1, an NADase homologous to mammalian SARM1, underlies p38 PMK-1 immune pathway activation in the *C. elegans* intestine (*Figure 6L*). Importantly, we demonstrate that physiologic stresses, both pathogen and non-pathogen, induce multimerization of TIR-1/SARM1 into visible puncta within intestinal epithelial cells. In vitro biochemical studies of purified protein and *C. elegans* genetic analyses



**Figure 6.** Sterol scarcity primes p38 PMK-1 immune defenses. (**A, B, C, D, E**) Images of the indicated transcriptional immune reporters under the indicated conditions. Scale bars in all images equal 200 μm. (**F**) A heat map compares the expression levels of the 184 genes that were both induced in wild-type animals during *P. aeruginosa* infection and differentially expressed (either induced or repressed) in *nhr-8(hd117)* mutants at baseline (each greater than two-fold, q < 0.05). To compare the expression of these genes in wild-type and *nhr-8(hd117)* mutants, we scaled the expression level in each condition by calculating a z-score for each row and performed hierarchical clustering, which identified two main clusters (Cluster I contains 139

*Figure 6 continued on next page*

*Figure 6 continued*

genes and Cluster II contains 45 genes). See also ***Supplementary file 5***. (**G**) Enrichment of p38 PMK-1-dependent genes in Cluster I and II genes is shown. (**H, I, J, K**) mRNA-seq data for the indicated genes from the experiment described in (**F**) showing scaled reads per base from three biological replicates. Error bars represent SEM. *equals q < 0.05 from RNA-seq analysis. (**L**) Model of p38 PMK-1 pathway activation during sterol scarcity and pathogen infection.

The online version of this article includes the following source data for figure 6:

**Source data 1.** *Figure 6H* mRNA-seq data for T24B8.5 from the conditions indicated showing scaled reads per base.

**Source data 2.** *Figure 6I* mRNA-seq data for *irg-4* from the conditions indicated showing scaled reads per base.

**Source data 3.** *Figure 6J* mRNA-seq data for *irg-5* from the conditions indicated showing scaled reads per base.

**Source data 4.** *Figure 6K* mRNA-seq data for K08D8.4 from the conditions indicated showing scaled reads per base.

revealed that multimerization and a phase transition of TIR-1/SARM1 engages its NADase activity to activate the p38 PMK-1 innate immune pathway and provide protection during bacterial infection.

Second, we show that cholesterol deficiency, as recapitulated in an *nhr-8* loss-of-function mutant, causes TIR-1/SARM1 to oligomerize into puncta in intestinal epithelial cells and activate the p38 PMK-1 immune pathway through its intrinsic NADase activity. *C. elegans* lacks the ability to synthesize cholesterol de novo and must acquire dietary sterols to support multiple aspects of organismal physiology (***Hieb and Rothstein, 1968***; ***Chitwood, 1999***; ***Shim et al., 2002***; ***Merris et al., 2003***; ***Yochem et al., 1999***; ***Otarigho and Aballay, 2020***; ***Cheong et al., 2011***). We found that priming of innate immune defenses during conditions of low cholesterol availability augments immune effector induction and leads to decreased accumulation of pathogen in the intestine during a subsequent bacterial infection. We therefore propose that immune activation in this manner represents a new adaptive response that allows a metazoan host to anticipate environmental threats during cholesterol deprivation, a time of relative susceptibility to infection.

Loring et al. demonstrated that human SARM1 aggregates and undergoes a phase transition to potentiate its intrinsic NAD⁺ glycohydrolase activity (***Loring et al., 2021***). These authors studied SARM1 in the context of neuronal degeneration as it was previously demonstrated that loss of mammalian SARM1 protects against axonal degeneration following neuronal injury (***Marion et al., 2019***; ***Henninger et al., 2016***; ***Geisler et al., 2016***). Treatment with non-physiologic concentrations of citrate, a molecule that induces protein precipitation, led to puncta formation of *C. elegans* TIR-1 in neurons, which was correlated with enhanced axonal degeneration (***Loring et al., 2021***). Moreover, previous studies showed that human SARM1 forms oligomers (***Bratkowski et al., 2020***; ***Jiang et al., 2020***) and that dimerization, at least, is required for TIR domain activity (***Gerdts et al., 2015***; ***Summers et al., 2016***; ***Zhao et al., 2019***). Thus, the demonstration here that *C. elegans* TIR-1/SARM1 aggregates in vivo in response to physiological stresses provides an important characterization of the mechanism inferred by Loring et al. and suggests that oligomerization favors the phase transition of TIR-1/SARM1 such that higher order structures are formed. Of note, the enzyme kinetics of TIR-1 are strikingly similar to mammalian SARM1 – the activity of both proteins is exquisitely dependent on precipitation. Thus, together, these studies demonstrate that a phase transition of TIR-1/SARM1 as a prerequisite for its NAD⁺ glycohydrolase activity is conserved across millions of years of evolution, occurs in multiple cell types, and is essential for diverse physiological processes, including intestinal immune regulation and axonal degeneration.

Currently, it is unknown how low cholesterol leads to the oligomerization and phase transition of *C. elegans* TIR-1/SARM1 and further studies are needed to characterize this mechanism. Cholesterol could be sensed to trigger this response, or low cholesterol could change the biophysical properties of the cell in a manner that favors oligomerization of TIR-1.

We demonstrate that cholesterol supplementation can fully rescue the enhanced susceptibility to pathogen-mediated killing in *nhr-8* loss-of-function mutants (***Figure 3M and N***). These data suggest that cholesterol is required for pathogen tolerance in *C. elegans*; however, the mechanism behind this observation is unknown. The p38 PMK-1 pathway provides protection during pathogen infection in the *nhr-8* mutant background (***Figure 4M***), suggesting that the enhanced susceptibility to pathogen-mediated killing in the *nhr-8* mutant is not secondary to aberrant activation of the p38 PMK-1 pathway. Cholesterol-deficient animals may therefore have general reductions in organismal fitness that cause vulnerability to pathogen infection and a reduction in lifespan.

The robust activation of the p38 PMK-1 pathway and the reduced pathogen burden in *nhr-8* mutants suggests that immune activation in this manner promotes pathogen resistance, which we propose is part of a mechanism to prime protective immune activation. It is also possible that activation of the p38 PMK-1 pathway during cholesterol deficiency could engage another protective host response that is unrelated to pathogen defense. We do not favor this hypothesis, however, considering the observed reduction in the pathogen burden of the *nhr-8* mutants (*Figure 3O*).

We previously demonstrated that a *C. elegans* nuclear hormone receptor, NHR-86, a member of a family of ligand-gated transcription factors, surveys the chemical environment to activate the expression of immune effectors (*Peterson et al., 2019*). Interestingly, NHR-86 targets immune effectors whose basal regulation requires the p38 PMK-1 immune pathway. However, NHR-86 functions independently of PMK-1 and modulates the transcription of these infection response genes directly. NHR-86 is a nematode homolog of the mammalian nuclear hormone receptor hepatocyte nuclear factor 4 (HNF4), a family of NHRs that expanded dramatically in *C. elegans* (259 HNF4 homologs are encoded in the *C. elegans* genome) (*Sluder et al., 1999*; *Sluder and Maina, 2001*). One potentially unifying hypothesis is that HNF4 homologs detect pathogen- or host-derived ligands that are associated with infection and activate pathogen-specific immune defenses. In this model, the p38 PMK-1 pathway functions as a rheostat that receives inputs from signals associated with potentially dangerous environmental conditions to prime host immune effector genes. Indeed, inputs from chemosensory neurons, bacterial density, tissue damage, and nucleotide metabolism also regulate the tonic level of p38 PMK-1 pathway activity (*Couillault et al., 2004*; *Pujol et al., 2008*; *Tecle et al., 2021*; *Foster et al., 2020a*; *Wu et al., 2019*; *Anderson and Pukkila-Worley, 2020*; *Styer et al., 2008*; *Cao and Aballay, 2016*; *Cao et al., 2017*; *Sun et al., 2011*), suggesting that p38 PMK-1 phosphorylation is adjusted to anticipate dangerous threats during periods of vulnerability. We hypothesize that mechanisms in *C. elegans* that detect specific pathogens can further augment immune effector induction.

## Materials and methods

**Key resources table**

| Reagent type (species) or resource | Designation | Source or reference | Identifiers | Additional information |
|---|---|---|---|---|
| Strain, strain background (*Escherichia coli*) | OP50 | *Caenorhabditis* Genetics Center | | |
| Strain, strain background (*Escherichia coli*) | HT115 | *Caenorhabditis* Genetics Center | | |
| Strain, strain background (*Escherichia coli*) | BL21 (DE3) | ThermoFisher Scientific | EC0114 | Chemically competent |
| Strain, strain background (*Escherichia coli*) | XL1-Blue | Agilent | 200,249 | Chemically competent |
| Strain, strain background (*Pseudomonas aeruginosa*) | UCBPP-PA14 | PMID:7604262 | | |
| Strain, strain background (*Caenorhabditis elegans*) | N2; wild-type | CGC, PMID:4366476 | WormBase ID: WBStrain00000001 | Laboratory reference strain/wild type |
| Strain, strain background (*Caenorhabditis elegans*) | KU25 | PMID:15116070/ | WormBase ID: WBStrain00024040 | Genotype: *pmk-1(km25) IV* |
| Strain, strain background (*Caenorhabditis elegans*) | AU3 | PMID:12142542 | WormBase ID: WBStrain00000259 | Genotype: *nsy-1(ag3) II* |
| Strain, strain background (*Caenorhabditis elegans*) | ZD101 | PMID:19837372 | WormBase ID: WBStrain00040806 | Genotype: *tir-1(qd4) III* |
| Strain, strain background (*Caenorhabditis elegans*) | AA968 | PMID:23931753 | | Genotype: *nhr-8(hd117) IV* |
| Strain, strain background (*Caenorhabditis elegans*) | AE501 | PMID:11516648 | WormBase ID: WBStrain00000059 | Genotype: *nhr-8(ok186) IV* |

*Continued on next page*

*Continued*

| Reagent type (species) or resource | Designation | Source or reference | Identifiers | Additional information |
|---|---|---|---|---|
| Strain, strain background (*Caenorhadbditis elegans*) | AU78 | PMID:19837372 | WormBase ID: WBStrain00000262 | Genotype: *agIs219* [T24B8.5p::*gfp*::*unc-54*–3'UTR; *ttx-3p*::*gfp*::*unc-54*–3'UTR] *III* |
| Strain, strain background (*Caenorhabditis elegans*) | AU307 | PMID:24875643 | | Genotype: *agIs44* [*irg-4p*::*gfp*::*unc-54*–3'UTR; *myo-2p*::*mCherry*] |
| Strain, strain background (*Caenorhabditis elegans*) | AY101 | PMID:20133945 | WormBase ID: WBStrain00000322 | Genotype: *acIs101* [p*DB09.1*(*irg-5p*::*gfp*); pRF4(*rol-6(su1006)*)] |
| Strain, strain background (*Caenorhabditis elegans*) | TJ356 | PMID:11747825 | WormBase ID: WBStrain00034892 | Genotype: *zIs356* [*daf-16p*::*daf-16a/b*::gfp + pRF4(*rol-6(su1006)*)] |
| Strain, strain background (*Caenorhabditis elegans*) | RPW278 | This study | | Genotype: *nhr-8(hd117);agIs219* |
| Strain, strain background (*Caenorhabditis elegans*) | RPW317 | This study | | Genotype: *tir-1(qd4);nhr-8(hd117)* |
| Strain, strain background (*Caenorhabditis elegans*) | RPW339 | This study | | Genotype: *tir-1(ums47[E788A]);agIs219* |
| Strain, strain background (*Caenorhabditis elegans*) | RPW369 | This study | | Genotype: *tir-1(ums54[ΔSAM]);agIs219* |
| Strain, strain background (*Caenorhabditis elegans*) | RPW374 | This study | | Genotype: *tir-1(ums55[G747P]);agIs219* |
| Strain, strain background (*Caenorhabditis elegans*) | RPW381 | This study | | Genotype: *tir-1(ums56[H833A]);agIs219* |
| Strain, strain background (*Caenorhabditis elegans*) | RPW386 | This study | | Genotype: *tir-1(ums57[tir-1::3xFLAG]);agIs219* |
| Strain, strain background (*Caenorhabditis elegans*) | RPW387 | This study | | Genotype: *tir-1(ums58[tir-1[E788A]::3xFLAG]);agIs219* |
| Strain, strain background (*Caenorhabditis elegans*) | RPW388 | This study | | Genotype: *tir-1 (ums59[tir-1[ΔSAM]::3xFLAG]);agIs219* |
| Strain, strain background (*Caenorhabditis elegans*) | RPW 389 | This study | | Genotype: *tir-1(ums60[tir-1[G747P]::3xFLAG]);agIs219* |
| Strain, strain background (*Caenorhabditis elegans*) | RPW403 | This study | | Genotype: *tir-1(ums63[tir-1::wrmScarlet])* |
| Strain, strain background (*Caenorhabditis elegans*) | RPW404 | This study | | Genotype: *nhr-8(hd117);tir-1(ums63[tir-1::wrmScarlet])* |
| Antibody | anti-total PMK-1 (rabbit polyclonal) | PMID:30668573 | | WB(1:1000) |
| Antibody | anti-Phospho-p38 MAPK (Thr180/Tyr182) (rabbit polyclonal) | Cell Signaling Technology | 9211 | WB(1:1,000) |
| Antibody | anti-FLAG (mouse monoclonal) | Sigma-Aldrich | F1804 | WB(1:1,000) |
| Antibody | anti-alpha-tubulin (mouse monoclonal) | Sigma-Aldrich | T5168 | WB(1:2,000) |
| Antibody | anti-mouse IgG-HRP (goat polyclonal) | Abcam | ab6789 | WB(1:10,000) |
| Antibody | anti-rabbit IgG-HRP (goat polyclonal) | Cell Signaling Technology | 7074 | WB(1:10,000) |
| Chemical compound, drug | Peptone | Gibco | 211820 | |
| Chemical compound, drug | Agar | Fisher | BP9744 | |

*Continued on next page*

*Continued*

| Reagent type (species) or resource | Designation | Source or reference | Identifiers | Additional information |
| --- | --- | --- | --- | --- |
| Chemical compound, drug | Cholesterol | Sigma Aldrich | C3045 | |
| Chemical compound, drug | Tri-Reagent | Sigma Aldrich | T9424 | |
| Chemical compound, drug | TERGITOL solution (Type NP-40) | Sigma Aldrich | NP40S | |
| Chemical compound, drug | RIPA Buffer | Cell Signaling Technology, Inc | 89900 | |
| Chemical compound, drug | Halt Protease and Phosphatase inhibitor | ThermoFisher Scientific | 78445 | |
| Chemical compound, drug | NuPAGE LDS sample buffer | ThermoFisher Scientific | NP0007 | |
| Chemical compound, drug | 1,6-hexanediol | Sigma Aldrich | 240117 | |
| Chemical compound, drug | Nicotinamide 1,$N^6$-ethenoadenine dinucleotide; ε-NAD | Sigma Aldrich | N2630 | |
| Chemical compound, drug | PEG 400 | Sigma Aldrich | 91893 | |
| Chemical compound, drug | PEG 1500 | Sigma Aldrich | 86101 | |
| Chemical compound, drug | PEG 3350 | Sigma Aldrich | 88276 | |
| Chemical compound, drug | PEG 8000 | Sigma Aldrich | 89510 | |
| Chemical compound, drug | Dextran | Fisher Scientific | ICN16011010 | |
| Chemical compound, drug | Sucrose | Sigma Aldrich | S0389 | |
| Chemical compound, drug | Glycerol | Sigma Aldrich | G5516 | |
| Chemical compound, drug | Sodium citrate | Sigma Aldrich | S4641 | |
| Chemical compound, drug | Kanamycin | Research Products International | K22000 | |
| Chemical compound, drug | IPTG | ThermoFisher Scientific | R0392 | |
| Chemical compound, drug | Pierce EDTA-free protease inhibitor mini tablets | ThermoFisher Scientific | A32955 | |
| Commercial assay or kit | iProof High-Fidelity DNA Polymerase | Bio-Rad Laboratories, Inc | 172–5301 | |
| Commercial assay or kit | iScript gDNA Clear cDNA Synthesis Kit | Bio-Rad Laboratories, Inc | 172–5034 | |
| Commercial assay or kit | iTaq Universal SYBR Green Supermix | Bio-Rad Laboratories, Inc | 1725120 | |
| Commercial assay or kit | DreamTaq Green PCR | ThermoFisher Scientific | K1081 | |
| Commercial assay or kit | DC protein assay | Bio-Rad Laboratories, Inc | 5000116 | |

*Continued*

| Reagent type (species) or resource | Designation | Source or reference | Identifiers | Additional information |
|---|---|---|---|---|
| Commercial assay or kit | NuPAGE 4%–12% BisTris gels | ThermoFisher Scientific | NP0321BOX | |
| Commercial assay or kit | NuPAGE 3%–8% TrisAcetate gels | ThermoFisher Scientific | EA0375BOX | |
| Commercial assay or kit | SuperSignal West Pico PLUS Chemiluminescent Substrate | Thermo Fisher Scientific | 34577 | |
| Commercial assay or kit | SuperSignal West Femto PLUS Chemiluminescent Substrate | Thermo Fisher Scientific | 34095 | |
| Commercial assay or kit | Strep-Tactin XT Superflow high-capacity resin | IBA Lifesciences | 2-4030-025 | Product discontinued; suitable replacement is Strep-Tactin XT 4Flow high capacity resin (2-5030-025) |
| Commercial assay or kit | Wizard Plus SV Minipreps DNA Purification System | Promega | A1460 | |
| Commercial assay or kit | TALON Metal Affinity Resin | Takara | 635,502 | |
| Peptide, recombinant protein | Lysozyme | Sigma Aldrich | L6876 | |
| Peptide, recombinant protein | *Dpn*I | NEB | R0176S | |
| Peptide, recombinant protein | ADP-ribosyl cyclase | Sigma Aldrich | A9106 | |
| Peptide, recombinant protein | SpCas9 Nuclease | IDT | 1081058 | |
| Recombinant DNA reagent | pET-30a(+) TIR (plasmid) | ***Loring et al., 2021*** | | Referred to as pET30a[+] Strep-ceTIR-HIS in reference |
| Software, algorithm | Fiji/imageJ | PMID:22743772 | | |
| Software, algorithm | OASIS 2 | PMID:27528229 | | |
| Software, algorithm | R Console (Version 3.5) | The R Foundation | | https://www.r-project.org/ |
| Software, algorithm | FastQC (Version 0.11.5) | https://www.bioinformatics.babraham.ac.uk/projects/fastqc/ | | |
| Software, algorithm | Kallisto (version 0.45.0) | PMID:27043002 | | |
| Software, algorithm | Sleuth (version 0.30.0) | PMID:28581496 | | |
| Software, algorithm | GSEA (version 4.1.0) | PMID:16199517 | | |
| Software, algorithm | pheatmap (version 1.0.12) | https://cran.r-project.org/web/packages/pheatmap/index.html | | |
| Software, algorithm | DAVID Bioinformatics database | PMID:19131956 | | |
| Software, algorithm | GraphPad Prism 9 | Graphpad | | https://www.graphpad.com/scientific-software/prism/ |

## *C. elegans* and bacterial strains

The previously published *C. elegans* strains used in this study were: N2 Bristol (***Brenner, 1974***), KU25 *pmk-1(km25)* (***Mizuno et al., 2004***), AU3 *nsy-1(ag3)* (***Kim et al., 2002***), ZD101 *tir-1(qd4)* (***Shivers et al., 2009***), AA968 *nhr-8(hd117)* (***Magner et al., 2013***), AE501 *nhr-8(ok186)* (***Lindblom et al., 2001***), AU78 *agIs219* [T24B8.5p::*gfp*::*unc-54*–3'UTR; *ttx-3*p::*gfp*::*unc-54*–3'UTR] (***Shivers et al., 2009***), AU306 *agIs44* [*irg-4*p::*gfp*::*unc-54*–3'UTR; *myo-2*p::mCherry] (***Pukkila-Worley et al., 2014***),

AY101 *acIs101* [p*DB09.1*(*irg-5*p::*gfp*); pRF4(*rol-6(su1006)*)] (*Bolz et al., 2010*), TJ356 *zIs356* [*daf-16*p::*daf-16a/b::gfp* + pRF4(*rol-6(su1006)*)] (*Henderson and Johnson, 2001*). The strains developed in this study were: RPW278 *nhr-8(hd117);agIs219*, RPW317 *tir-1(qd4);nhr-8(hd117)*, RPW339 *tir-1(ums47[E788A]);agIs219*, RPW369 *tir-1(ums54[ΔSAM]);agIs219*, RPW374 *tir-1(ums55[G747P]);agIs219*, RPW381 *tir-1(ums56[H833A]);agIs219*, RPW386 *tir-1(ums57[tir-1::3xFLAG]);agIs219*, RPW387 *tir-1(ums58[tir-1[E788A]::3xFLAG]);agIs219*, RPW388 *tir-1 (ums59[tir-1[ΔSAM]::3xFLAG]);agIs219*, RPW 389 *tir-1(ums60[tir-1[G747P]::3xFLAG]);agIs219*, RPW403 *tir-1(ums63[tir-1::wrmScarlet])*, RPW404 *nhr-8(hd117);tir-1(ums63[tir-1::wrmScarlet])*. Bacteria used in this study are *Escherichia coli* OP50, *E. coli* DH5α, *E. coli* HT115(DE3), and *Pseudomonas aeruginosa* strain PA14 (*Rahme et al., 1995*).

## *C. elegans* growth conditions and lipid supplementation

*C. elegans* strains were maintained on standard nematode growth medium (NGM) plates (0.25% bacto peptone, 0.3% sodium chloride, 1.7% agar [Fisher], 5 µg/mL cholesterol [Sigma-Aldrich, BioReagent grade], 25 mM potassium phosphate pH 6.0, 1 mM magnesium sulfate, 1 mM calcium chloride) with *E. coli* OP50 as a food source, as described (*Brenner, 1974*). For low-cholesterol medium (0 µg/mL cholesterol), NGM was prepared without cholesterol supplementation, while 0.1% ethanol was added to maintain an equivalent ethanol concentration. For high-cholesterol medium, cholesterol was dissolved in ethanol at 20 mg/mL and added to NGM at a final concentration of 80 µg/mL immediately prior to pouring plates. For all assays with high-cholesterol medium, NGM containing 0.4% ethanol and 5 µg/mL cholesterol were used as control plates. Cholesterol solubilization assays were performed by supplementing NGM containing 5 µg/mL cholesterol with either 0.1% Tergitol (Sigma-Aldrich) or 0.1% Triton X-100 (Sigma-Aldrich). For assays using media solidified with agarose, NGM plates were prepared with 1.7% Ultrapure agarose (ThermoFisher Scientific) in place of agar. All fatty acids were purchased from Nu-Check-Prep Inc, and supplementation performed as previously described with modification (*Anderson et al., 2019*; *Nandakumar and Tan, 2008*). Fatty acids were dissolved in 50% ethanol and added at a final concentration of 1 mM to NGM agarose containing 0 µg/mL cholesterol and 0.1% Tergitol immediately prior to plate pouring. Prior to all assays, plates supplemented with lipids and control plates were seeded with *E. coli* OP50 and grown for 24 hr at room temperature. Assays were performed by picking 10–20 gravid adult animals to either lipid-supplemented or matched control plates. Animals were maintained on the plates for 14 hr at 20 °C, after which they were removed. Eggs laid on the plate were allowed to hatch and develop to the L4 stage at 20 °C. For low-cholesterol assays, animals were grown for two generations on NGM containing 0 µg/mL cholesterol.

## *C. elegans* strain construction

CRISPR/Cas9 was used to generate *tir-1* mutants in both wild-type and TIR-1::3xFLAG backgrounds, as described (*Dokshin et al., 2018*; *Ghanta and Mello, 2020*). All CRISPR/Cas9 reagents were purchased from Integrated DNA Technologies. Target guide sequences were selected using the CHOPCHOP web tool (*Labun et al., 2019*). ssODN and dsDNA repair templates contained indicated edits, deletions or insertions with 35 bp flanking homology arms. crRNA guide and ssODN sequences are listed in *Supplementary file 6*. For wrmScarlet dsDNA repair template, wrmScarlet was PCR amplified with 35 bp flanking homology arms. PCR was gel purified, diluted to 100 ng/µL, melted and reannealed using a thermal cycler (95 °C – 2 min; 85 °C – 10 s, 75 °C – 10 s, 65 °C – 10 s, 55 °C – 10 s, 45 °C – 10 s, 35 °C – 10 s, 25 °C – 10 s, 4 °C – hold. Ramp down 1 °C per s), and used immediately for injection. A mixture of 0.25 µg/µL Cas9, 0.1 µg/µL tracrRNA and 0.056 µg/µL crRNA were incubated for 15 min at 37 °C. 0.11 µg/µL ssODN or 25 ng/µL dsDNA and 40 ng/µL pRF4(*rol-6(su1006)*) plasmid were added to the mixture, centrifuged, and microinjected into young adult animals carrying the *agIs219* transgene, *tir-1(ums57[tir-1::3xFLAG]);agIs219*, or N2. The F1 progeny were screened for Rol phenotypes 3–4 days after injection and then for indicated edits using PCR and Sanger sequencing. Primer sequences used for genotyping are listed in *Supplementary file 6*.

## Feeding RNAi

Knockdown of target genes was performed by feeding *C. elegans E. coli* HT115 expressing dsRNA targeting the gene of interest, as previously described with modification (*Fire et al., 1998*; *Timmons et al., 2001*; *Conte et al., 2015*). In brief, HT115 bacteria expressing dsRNA targeting genes of

interest were grown in Lysogeny broth (LB) Lennox medium containing 50 µg/mL ampicillin and 15 µg/mL tetracycline overnight with shaking (250 rpm) at 37 °C. Overnight cultures were seeded onto NGM containing 5 mM IPTG and 50 µg/mL carbenicillin and incubated at 37 °C for 16 hr, after which synchronized L1 animals were transferred to bacterial lawns and allowed to grow until the L4 stage.

## *C. elegans* bacterial infection and colonization assays

"Slow killing" *P. aeruginosa* infection experiments were performed as previously described (**Tan et al., 1999**; **Foster et al., 2020b**). Wild-type is either N2 or *agIs219*. In brief, a single colony of *P. aeruginosa* PA14 was inoculated into 3 mL of LB medium and grown with shaking (250 rpm) at 37 °C for 14 hr. Ten µL of overnight culture was spread onto 35 mm petri dishes containing 4 mL slow killing agar (0.35% peptone, 0.3% sodium chloride, 1.7% agar, 5 µg/mL cholesterol, 25 mM potassium phosphate pH 6.0, 1 mM magnesium sulfate, 1 mM calcium chloride). Plates were incubated for 24 hr at 37 °C and for approximately 24 hr at 25 °C. Immediately prior to starting the assay, 0.1 mg/mL 5-fluorodeoxyuridine (FUDR) was added on top of the agar to prevent progeny from hatching. Animals used in all assays were grown at 20 °C with specified growth conditions. For assays involving high cholesterol or nonionic detergents, slow-killing agar plates were prepared with either 80 µg/mL cholesterol (Sigma-Aldrich), 0.1% Tergitol (Sigma-Aldrich), or 0.1% Triton X-100 (Sigma-Aldrich). For experiments using plates containing 80 µg/mL cholesterol, matched control plates containing the equivalent ethanol concentration (0.4%) and 5 µg/mL cholesterol were prepared. All pathogenesis and lifespan assays are representative of three biological replicates. Sample sizes, mean lifespan, and p values for all trials are shown in **Supplementary file 4**.

CFU of *P. aeruginosa* were quantified in the intestine of *C. elegans* as previously described with modifications (**Foster et al., 2020b**; **Singh and Aballay, 2019**). Briefly, *C. elegans* animals were exposed to lawns of *P. aeruginosa*, which were prepared as previously described, for 24 hr. Animals were then picked to NGM plates lacking bacteria and incubated for 10 min to remove external *P. aeruginosa*. Animals were then transferred to a second NGM plate, after which 10–11 animals per replicate were collected, washed with M9 buffer containing 25 mM tetramisole (Sigma-Aldrich) and 0.01% Triton X-100 (Sigma-Aldrich), and ground with 1.0 mm silicon carbide beads (BioSpec Products). *P. aeruginosa* CFUs were quantified from serial dilutions of the lysate grown on LB agar.

## Gene expression analysis and bioinformatics

A total of 2000 synchronized L1 stage *C. elegans* of the indicated genotypes were grown to the L4 stage and harvested by washing with M9. For expression analysis of *C. elegans* genes during *P. aeruginosa* infection, animals at the L4 stage animals were transferred by washing to plates containing *E. coli* OP50 or *P. aeruginosa* PA14 lawns. Animals were exposed for four hours and subsequently harvested by washing with M9. RNA was isolated using TriReagent (Sigma-Aldrich), column purified (Qiagen), and analyzed by 100 bp paired-end mRNA-sequencing using the BGISEQ-500 platform (BGIAmericasCorp) with > 20 million reads per sample. Raw fastq reads were evaluated by FastQC (version 0.11.5), clean reads were aligned to the *C. elegans* reference genome (WBcel235) and quantified using Kallisto (version 0.45.0) (**Bray et al., 2016**). Differentially expressed genes were identified using Sleuth (version 0.30.0) (**Pimentel et al., 2017**). Pearson correlation statistical analysis was performed using Prism 9.0. Innate immune genes were identified using DAVID Bioinformatics database biological process gene ontology (GO) term innate immune response (**Huang et al., 2009**). Heatmaps of differentially expressed genes were generated using pheatmap (version 1.0.12). Gene set enrichment analysis of RNA-seq was performed using GSEA (version 4.1.0) (**Subramanian et al., 2005**) with a custom gene set database (**Supplementary file 7**) containing p38 PMK-1-dependent genes generated by analyzing previously published RNA-seq of uninfected *pmk-1(km25)* animals (**Fletcher et al., 2019**) with the RNA-seq data analysis pipeline described above. Differential gene expression was defined as a fold change (FC) versus wild-type greater than two and q less than 0.01.

For the qRT-PCR studies, RNA was reverse transcribed to cDNA using the iScript cDNA Synthesis Kit (Bio-Rad Laboratories, Inc), amplified and detected using Syber Green (Bio-Rad Laboratories, Inc) and a CFX384 machine (Bio-Rad Laboratories, Inc). The sequences of primers that were designed for this study are presented in **Supplementary file 6**. Other primers were previously published (**Troemel et al., 2006**; **Taubert et al., 2008**; **Richardson et al., 2010**; **Estes et al., 2010**). All values were

normalized against the geometric mean of the control genes *snb-1* and *act-3*. Fold change was calculated using the Pfaffl method (**Pfaffl, 2001**).

## Immunoblot analyses

Protein lysates from 2000 *C. elegans* grown to the L4 larval stage on *E. coli* OP50 on NGM agar were prepared as previously described with modification (**Peterson et al., 2019**; **Cheesman et al., 2016**). Harvested animals were washed twice with M9 buffer and resuspended in RIPA Buffer (Cell Signaling Technology, Inc) containing 1 x Halt Protease and Phosphatase inhibitor (ThermoFisher Scientific). Samples were lysed using a teflon homogenizer, centrifuged, and protein was quantified from the supernatant of each sample using the DC protein assay (Bio-Rad Laboratories, Inc). NuPAGE LDS sample buffer (ThermoFisher Scientific) was added to a concentration of 1 X, and 12.5–30 µg of total protein from each sample was resolved on NuPAGE 4–12% BisTris (Phospho-PMK-1 and Total-PMK-1) or NuPAGE 3–8% TrisAcetate (TIR-1::3xFLAG) gels (ThermoFisher Scientific), transferred to nitrocellulose membranes using a Trans-Blot Turbo Transfer System (Bio-Rad Laboratories, Inc), blocked with 5% milk powder in TBST and probed with a 1:1,000 dilution of an antibody that recognizes the doubly-phosphorylated TGY motif of PMK-1 (Cell Signaling Technology, #9211), a previously characterized total PMK-1 antibody (**Peterson et al., 2019**), a monoclonal mouse anti-FLAG antibody (Sigma-Aldrich, M2), or a monoclonal mouse anti-alpha-tubulin antibody (Sigma-Aldrich, Clone B-5-1-2). Horseradish peroxidase (HRP)-conjugated anti-rabbit (Cell Signaling Technology, #7074) and anti-mouse IgG secondary antibodies (Abcam, #ab6789) were diluted 1:10,000 and used to detect the primary antibodies following the addition of ECL reagents (Thermo Fisher Scientific, Inc), which were visualized using a BioRad ChemiDoc MP Imaging System. The band intensities were quantified using Fiji/ImageJ, and the ratio of active phosphorylated PMK-1 to total PMK-1 was calculated.

## TIR-1::wrmScarlet puncta visualization

For TIR-1::wrmScarlet puncta visualization, all TIR-1::wrmScarlet expressing animals were depleted of autofluorescent gut granules by exposure to *glo-3(RNAi)* from the L1 stage prior to all experiments. *P. aeruginosa* was prepared as described for bacterial infection and colonization assays. L4 animals expressing TIR-1::wrmScarlet were transferred to either OP50 or *P. aeruginosa* containing plates for 24 hr. For imaging, animals were transferred to 2% agarose pads, paralyzed with 300 mM sodium azide, and imaged with a 63 x oil immersion lens. For Tergitol experiments, animals were grown on RNAi plates containing 0.1% Tergitol and the respective RNAi strains and imaged at the L4 stage.

## Microscopy

Nematodes were mounted onto agar pads, paralyzed with 10 mM levamisole (Sigma) or 300 mM sodium azide and photographed using a Zeiss AXIO Imager Z2 microscope with a Zeiss Axiocam 506mono camera and Zen 2.3 (Zeiss) software.

## Microscopy image analysis

Image processing and analysis were performed using Fiji image analysis software. In each panel, the exposure time and processing for all images were identical. To quantify T24B8.5p::gfp fluorescence intensity, the DIC channel was used to identify and select the intestine of each animal using the Freehand Tool. The integrated density, which is the product of the area and mean gray value, was then calculated for the green channel of each selected region using the Measurement function. To quantify TIR-1::wrmScarlet puncta, the last posterior pair of intestinal epithelial cells were identified in the DIC channel and selected using the Freehand Tool. We then filtered the red and green channel images using Gaussian Smoothing (sigma radius = 1). The Find Maxima tool was then used to identify puncta in the red channel within the selected area. The Find Maxima settings used for the analysis of *P. aeruginosa* puncta were prominence: 5, strict: no, exclude edge maxima: no. The settings for the *nhr-8(hd117)* and Tergitol experiments were prominence: 15, strict: no, exclude edge maxima: no. We then overlayed the location of the red channel puncta in the green channel and determined if there was an autofluorescent puncta in this exact location. TIR-1::wrmScarlet puncta were those that were present in the red, but not the green, channel. Importantly, we only reported puncta that were positively identified by the image analysis software.

## TIR-1 TIR domain expression and purification

The recombinant *C. elegans* TIR-1 TIR domain (TIR) was expressed in bacteria as previously described (*Loring et al., 2020*). Briefly, the TIR domain cloned into the pET-30a(+) vector was transformed into chemically competent *E. coli* BL21(DE3) cells and maintained as a glycerol stock at –80 °C. An inoculation loop was used to transfer the transformed bacteria into 5 mL of LB media with 50 μg/mL (final concentration) of kanamycin, and the culture was grown overnight at 37 °C while rotating. The next day, the cultures were diluted 1:400 in LB media with 50 μg/mL (final concentration) of kanamycin and grown at 37 °C while shaking at 215 rpm until an $OD_{600}$ of 0.7–0.8 was reached. After cooling, 50 μM IPTG (final concentration) was added to the culture to induce protein expression. The incubator temperature was decreased to 16 °C, and cells were incubated for an additional 16–18 hr. Bacterial cells were collected by centrifugation at 3000 *x g* for 15 min at 4 °C, flash frozen in liquid nitrogen, and stored at –80 °C until purification.

For purification, bacterial pellets were thawed on ice and then resuspended in Lysis Buffer (50 mM Tris•HCl pH 7.0, 300 mM NaCl, 10% (w/v) glycerol, 0.001% Tween 20) with Pierce EDTA-free protease inhibitor mini tablets (ThermoFisher Scientific). The resuspension was incubated with 100 μg/mL lysozyme for 10 min at 4 °C and sonicated with a Fisher Scientific Sonic Dismembrator sonicator (FB-705) in 50 mL batches at an amplitude of 30 for 20 s, pulsing for 1 sec on and 1 s off, followed by a delay period of 20 s for a series of 12 cycles. Crude lysate was clarified at 21,000 *x g* for 25 min at 4 °C, at which point the supernatant was applied to pre-equilibrated Strep-Tactin XT Superflow high-capacity resin (IBA Lifesciences) and allowed to enter the column by gravity flow; the Strep-Tactin resin had been equilibrated in Strep Wash Buffer (50 mM Tris•HCl pH 7.0, 300 mM NaCl). The column was washed with 30 column volumes of Strep Wash Buffer, and the protein was eluted with 25 column volumes of Strep Elution Buffer (Strep Wash Buffer with 50 mM biotin). The protein eluted from the Strep-Tactin column was then applied to pre-equilibrated TALON Metal Affinity Resin (Takara) and allowed to enter the column by gravity flow; the TALON resin was equilibrated in His Wash 1 (50 mM Tris•HCl pH 7.0, 150 mM NaCl, 5 mM imidazole). A series of 15 column volume washes were applied (His Wash 1; His Wash 2: 50 mM Tris•HCl pH 7.0, 150 NaCl, 10 mM imidazole), and the protein was eluted in 20 column volumes of His Elution Buffer (50 mM Tris•HCl pH 7.0, 150 mM NaCl, 150 mM imidazole). The eluted protein was dialyzed overnight in Dialysis Buffer (50 mM Tris•HCl, pH 7.0, 150 mM NaCl). The next day, the protein was concentrated using a 10,000 NMWL Amicon Ultra-15 Centrifugal Filter Unit at 4 °C, and the protein concentration was determined by the Bradford assay. TIR was flash frozen in liquid nitrogen and stored at –80 °C in 25 μL aliquots.

## Fluorescent NADase assay

Nicotinamide 1,$N^6$-ethenoadenine dinucleotide (ε-NAD, Sigma-Aldrich) is a fluorescent analog of $NAD^+$ and was utilized in kinetic assays as a TIR-1 substrate. TIR-1 cleaves the nicotinamide moiety from ε-NAD to release nicotinamide and etheno-ADPR (ε-ADPR), which fluoresces ($\lambda_{ex}$ = 330 nm, $\lambda_{em}$ = 405 nm). Enzymatic activity was assayed in Assay Buffer (50 mM Tris pH 8.0, 150 mM NaCl; final concentration) using Corning 96–well Half Area Black Flat Bottom Polystyrene NBS Microplates for a final reaction volume of 60 μL, or Corning 384-well Low Volume Black Round Bottom Polystyrene NBS Microplates for a final volume of 20 μL; reactions were initiated by the addition of ε-NAD. ε-ADPR fluorescence intensity readings were taken in real time every 15 s for 15–30 min using Wallac EnVision Manager Software and a PerkinElmer EnVision 2,104 Multilabel Reader. Fluorescence intensity readings ($\lambda_{ex}$ = 330 nm, $\lambda_{em}$ = 405 nm) were converted to [ε-ADPR] with an ε-ADPR standard curve, which was produced by incubating fixed concentrations (0–400 μM) of ε-NAD with excess ADP-ribosyl cyclase and plotting the peak fluorescence intensity values against [ε-ADPR]. The activity was linear with respect to time under all conditions tested.

## Effect of crowding agents and sodium citrate on the activity of the TIR domain

The enzymatic activity of the TIR domain was evaluated using the Fluorescent Assay described above. First, the enzyme concentration dependence was determined. For purified protein, the enzyme (0–32.5 μM; final concentration) was added to Assay Buffer in duplicate, briefly incubated at room temperature for 10 min, and the reaction was initiated with 1 mM ε-NAD. Fluorescence intensity was monitored in real time every 15 s for 15 min.

To evaluate the effect of crowding agents and sodium citrate on TIR activity, stock solutions of the additives were made. 50% (w/v) solutions of PEGs 8000, 3500, 1500, and 400, as well as 60% (w/v) solutions of dextran, sucrose, and glycerol were prepared and filtered. Initially, the concentration dependence of TIR in the presence of PEG 3350 was determined. TIR (0–15 µM; final concentration) was added to Assay Buffer with 25% PEG 3350 (final concentration) in duplicate, incubated for 10 min at room temperature, initiated with 1 mM ε-NAD, and monitored every 15 sec for 15 min. The concentration dependence of TIR in 25% PEG 3350 was used to establish that 2.5 µM TIR could be used to enable robust kinetic analyses. Next, the effect of viscogens on TIR activity was evaluated by adding 2.5 µM TIR (final concentration) in duplicate to Assay Buffer with or without 25% w/v of the viscogens (final concentration). Following a brief 10-min incubation period, the reaction was initiated with 1 mM ε-NAD and monitored every 15 s for 20 min. The dose response of 2.5 µM TIR to the viscogens (0, 10, 20, and 30% final concentrations) was evaluated in the same manner and monitored every 15 s for 15 min.

Sodium citrate was prepared as a stock solution of 2 M and filtered. The concentration dependence of TIR in the presence of 500 mM sodium citrate was determined. TIR (0–15 µM; final concentration) was added to Assay Buffer with 500 mM citrate (final concentration) in duplicate, incubated for 10 min at room temperature, and then the reaction was initiated with 1 mM ε-NAD and monitored every 15 s for 15 min. Additionally, the dose dependence of sodium citrate was determined in duplicate. 2.5 µM TIR was added to Assay Buffer with (0–1000 mM) sodium citrate and briefly incubated at room temperature for 10 min. The reaction was initiated with 1 mM ε-NAD, and fluorescence was monitored every 15 s for 15 min.

In all cases, the fluorescence intensity was converted to ε-ADPR using the ε-ADPR standard curve described above. Slopes of the progress curves yielded the velocities of the reactions, which were plotted in GraphPad Prism.

## Effect of PEG 3350 and sodium citrate on steady-state kinetics

Steady-state kinetic reactions were carried out in Assay buffer with either PEG 3350 (0%–25%; final concentration) or sodium citrate (0–1000 mM; final concentration); a constant concentration of 2.5 µM TIR was used in these assays. Reaction components were mixed in duplicate and incubated at room temperature for 10 min before initiating the reaction with ε-NAD (0–4000 µM, final concentration). Fluorescence intensity was monitored every 15 s for 15 min. Using the ε-ADPR standard curve, the fluorescence was converted to [ε-ADPR]. The velocity of the reactions was calculated from the slope of the progress curve at each ε-NAD concentration and plotted in GraphPad Prism. Kinetic parameters were determined by fitting these velocities to the Michaelis-Menten equation (Eq. 1) at each PEG 3350 or sodium citrate concentration. $K_m$, $k_{cat}$, and $k_{cat}/K_m$ values were plotted against PEG 3350 or sodium citrate concentration.

$$v = \frac{V_{max}[S]}{K_m + [S]},$$ (1)

where $V_{max}$ is the maximum velocity, $[S]$ is the substrate concentration, and $K_m$ is the substrate concentration at half the maximum velocity.

## The TIR domain precipitates in PEG3350 and sodium citrate

Five µM TIR was incubated in Assay Buffer with PEG 3350 (0, 10, 17.5, and 25%; final concentration) or sodium citrate (0, 125, 250, 500, 750, and 1000 mM; final concentration) in duplicate at room temperature for 15 min. The precentrifugation control was removed, and the remainder of the sample was centrifuged at 17,000 x g at 4 °C for 10 min. Following centrifugation, the supernatant was separated from the pellet, and the pellet was resuspended in Assay Buffer with the respective concentration of PEG 3350 or sodium citrate. Samples were diluted 1:2 with gel loading buffer and run on an SDS-PAGE gel. Protein bands were stained by Coomassie and visualized on a BioRad Gel Doc EZ Gel Documentation System with Image Lab Software. Representative images are shown.

Following resuspension of the pellet, all fractions were analyzed in the Fluorescent Assay. After the samples were aliquoted into the assay plate in duplicate, the enzymatic reaction was initiated with 1 mM ε-NAD and monitored every 15 s for 15 min. Fluorescence intensity was converted to ε-ADPR

concentration using the ε-ADPR standard curve. Slopes of the progress curves yielded the velocities of the reactions, which were plotted in GraphPad Prism.

## Effect of 1,6-hexanediol on TIR domain activity and precipitation

To evaluate the effect of 1,6-hexanediol on TIR activity, the Fluorescent Assay was performed in triplicate in the absence and presence of PEG 3350 or sodium citrate. For pure protein, 35 µM TIR (final concentration) was incubated in Assay Buffer at room temperature for 10 min with or without 2% 1,6-hexanediol (final concentration). For PEG 3350 or citrate, 2.5 µM TIR was incubated in Assay Buffer with 25% PEG 3350 or 500 mM sodium citrate (final concentrations) for 10 min at room temperature with or without 2% 1,6-hexanediol. The reactions were initiated with 1 mM ε-NAD and monitored every 15 s for 15 min. Using the ε-ADPR standard curve, fluorescence was converted to [ε-ADPR], and the reaction velocities (i.e. slopes of the progress curves) were obtained. Velocities were normalized for enzyme concentration, and these normalized velocities were plotted in GraphPad Prism.

To determine whether 1,6-hexanediol can alter TIR precipitation, 1,6-hexanediol was added either before or after TIR precipitation in duplicate. To assess whether 1,6-hexanediol disrupts TIR precipitation, TIR precipitates were formed first by incubating 10 µM TIR (final concentration) with 25% PEG3350 or 500 mM sodium citrate (final concentration) at room temperature for 15 min. 1,6-hexanediol (0, 1, or 2%; final concentrations) was added to the mixture and incubated at room temperature for an additional 10 min. The precentrifugation control was removed, and the remaining mixture was centrifuged at 21,000 x g for 10 min at 4 °C. Supernatant fractions were removed, and the pellet was resuspended in Assay Buffer with the respective additive and concentration of 1,6-hexanediol. All fractions were run on an SDS-PAGE and stained with Coomassie Blue, and gels were imaged on a BioRad Gel Doc EZ Gel Documentation System with Image Lab Software. Next, we determined if 1,6-hexanediol could prevent TIR precipitation. 10 µM TIR was incubated in Assay Buffer with 1,6-hexanediol (0, 1, or 2%; final concentration) for 15 min at room temperature. 25% PEG 3350 or 500 mM sodium citrate was added to the TIR-buffer-hexanediol mixture and incubated further for 10 min. Controls were removed and the samples were centrifuged at 21,000 x g for 10 min at 4 °C. As before, supernatant fractions were removed, and pellet fractions were resuspended in Assay Buffer with respective additives and hexanediol concentrations. Fractions were analyzed by SDS-PAGE/Coomassie staining; representative images are shown.

## Phase transition reversibility

To evaluate the reversibility of the TIR phase transition, 5 µM of TIR was mixed with Assay Buffer and 25% PEG 3350 or 500 mM sodium citrate in duplicate (final concentrations). Following a 15-min incubation period at room temperature, precentrifugation controls were removed and the sample remaining was centrifuged at 17,000 x g for 10 min at 4 °C. Supernatant fractions were separated from the pellet, which was resuspended in either Assay Buffer alone or Assay Buffer with respective additive. All fractions were analyzed for enzymatic activity in the Fluorescent Assay. Briefly, the precentrifugation, supernatant, and pellet fractions were aliquoted into the assay plates in duplicate and the reaction was initiated with 1 mM ε-NAD. Fluorescence was converted to [ε-ADPR] concentration with the ε-ADPR curve to yield the progress curves. The velocity of the reactions was taken as the slope of the line and the velocities were plotted in GraphPad Prism.

To validate the kinetic data, 10 µM TIR (final concentration) was incubated in Assay Buffer with either 25% PEG 3350 or 500 mM sodium citrate (final concentration) for 15 min at room temperature; control samples were incubated in Assay Buffer only. Samples were centrifuged at 21,000 x g for 10 min at 4 °C, after which the supernatant was separated from the pellet. The pellets from samples initially prepared with additives were resuspended in either Assay Buffer alone or Assay Buffer plus the respective additive; this was not necessary for the sample initially prepared without additive since the protein is primarily located in the supernatant in this case. After removing another control sample (Additive lanes on gel), the resuspended samples were centrifuged again at 21,000 x g for 10 min at 4 °C. As before, the supernatant was removed, and the pellet was resuspended in Assay Buffer with the respective additive. All samples were analyzed for protein content on an SDS-PAGE gel and stained with Coomassie Blue.

## Effect of pH on TIR domain precipitation and kinetics

To determine the effect of pH on TIR precipitation, the experiments described above were carried out in duplicate at pH values from 4.5 to 9.0. Briefly, 5 µM TIR (final concentration) was mixed with Assay Buffer (50 mM buffer; 150 NaCl) with and without 25% PEG 3350 (final concentration); for pH 4.5–5, sodium acetate buffer was used; for pH 5.5–6.5, MES was used; for pH 7–9, Tris was used. The samples were incubated for 15 min at ambient temperature, at which point the samples were centrifuged at 21,000 x g for 10 min at 4 °C. The supernatant was removed, and the pellet was resuspended in either buffer alone or buffer with 25% PEG 3350 (final concentration); the presence or absence of 25% PEG 3350 and the buffer identity of the resuspension solution corresponded to the initial sample preparation. To neutralize the buffer, 10 µL of 1 M Tris (pH 6.8) was added to each sample before running on an SDS-PAGE gel and staining with Coomassie blue. Images of the gels were obtained on the BioRad Gel Doc EZ Gel Documentation System with Image Lab Software. Representative images are shown. ImageJ was used to quantify the bands, which were plotted in GraphPad Prism.

Steady-state kinetic analyses were also performed at each pH in 25% PEG 3350. A constant concentration of 2.5 µM TIR was used in these assays. Reaction components were mixed in quadruplicate and incubated at room temperature for 10 min before initiating the reaction with 0–2000 µM of ε-NAD (final concentration). Fluorescence intensity was monitored every 15 s for 15 min. Using the ε-ADPR standard curve, the fluorescence was converted to [ε-ADPR]. The velocity of the reactions was calculated from the slope of the progress curve at each ε-NAD concentration. Kinetic parameters were determined by fitting these velocities to the Michaelis-Menten equation (Eq. 1) at each pH. The log of $K_m$, $k_{cat}$, and $k_{cat}/K_m$ values were determined and plotted in GraphPad Prism.

## Negative stain electron microscopy

Negative stain EM on TIR (270 µg/mL) was performed in Assay Buffer (50 mM Tris, pH 8; 150 mM NaCl) with or without 500 mM sodium citrate. Samples were applied to glow-discharged, carbon- and formvar-coated copper grids and allowed to sit for 1 min and 30 s before blotting excess liquid away. 1% uranyl acetate was used to fix the samples before imaging. Samples were imaged on an FEI Tecnai Spirit 12 microscope. The diameters of particles in the samples with citrate were analyzed in ImageJ.

## TIR domain mutants

TIR domain mutants were made using PCR-based methods. The pET30a + TIR-1 TIR domain construct was used as a template (0.2–2 ng/µL; final concentration). The manufacturer's protocol for iProof High-Fidelity DNA Polymerase (Bio-Rad Laboratories, Inc) was followed using iProof HF buffer supplemented with 3% DMSO. 50 µL reaction volumes were used in the following protocol: initial denaturation for 3 min at 98 °C, denaturation for 45 s at 98 °C, annealing for 1:30 min at 45°C–72°C, extension for 6 min at 72 °C, final extension for 10 min at 72 °C. Denaturation, annealing, and extension steps were repeated 30 times. The following day, *Dpn*I was added to the PCR reactions and incubated at 37 °C for 2 hr to digest the template DNA. The digest was transformed into chemically competent *E. coli* XL1-Blue cells. Transformants were grown overnight in LB media with 50 µg/mL kanamycin and mini-prepped (Promega). Mutagenesis was validated by Sanger sequencing (Genewiz). The mutants were expressed and purified as described for WT TIR.

Steady-state kinetic analyses of the mutants were carried out in Assay Buffer with 25% PEG 3350 or 500 mM sodium citrate with 2.5 µM of the mutants (final concentrations). Reaction components were mixed in triplicate and incubated at room temperature for 10 min before initiating the reaction with 0–2000 µM of ε-NAD for PEG 3350 or 0–4000 µM for citrate. Fluorescence intensity was monitored every 15 s for 17.5 min. Using the ε-ADPR standard curve, the fluorescence was converted to [ε-ADPR]. The velocity of the reactions was calculated from the slope of the progress curve at each ε-NAD concentration and plotted in GraphPad Prism. Kinetic parameters were determined by fitting these velocities to the Michaelis-Menten equation (Eq. 1) for each mutant.

To evaluate the precipitation capacity of TIR mutants, 10 µM (WT, G747P, E788Q, and H833A) or 3 µM (WT, E788A) of the enzyme was mixed with Assay Buffer (50 mM Tris, pH 8.0; 150 NaCl) with and without 25% PEG 3350 or 500 mM sodium citrate (final concentrations). Precipitation of the TIR$^{E788A}$ mutant was evaluated at 3 µM due to low yields of the protein; WT TIR at 3 µM was included as the proper control. The samples were incubated for 15 min at ambient temperature, at which point the samples were centrifuged at 21,000 x g for 10 min at 4 °C. The supernatant was removed, and the

pellet was resuspended buffer with the respective additive (final concentration). Samples were diluted 1:1 with water before running on an SDS-PAGE gel. Stain-free images of the gels were obtained on the BioRad Gel Doc EZ Gel Documentation System with Image Lab Software. Representative images are shown. ImageJ was used to quantify the bands, which were plotted in GraphPad Prism.

## Statistical analyses

Differences in the survival of *C. elegans* in the *P. aeruginosa* pathogenesis assays were determined with the log-rank test after survival curves were estimated for each group with the Kaplan-Meier method. OASIS two was used for these statistical analyses (*Han et al., 2016*). qRT-PCR studies, intestinal CFU quantification, western blot band intensity quantification, TIR NADase activity, and TIR protein precipitation are presented as the mean ± standard error of the mean. Statistical hypothesis testing was performed with Prism 9 (GraphPad Software) using methods indicated in the figure legends. Sample sizes, mean lifespan, and p-values for all trials are shown in *Supplementary file 4*.

## Acknowledgements

The authors thank Alexandra Byrne, Victoria Czech, Lauren O'Connor, and Heather Loring for helpful discussions. The authors are also grateful to Melanie Trombly and Merin MacDonald for critical reading of the manuscript. This research was supported by R01 AI130289 (to RPW), R01 AI159159 (to RPW), R21 AI163430 (to RPW), an Innovator Award from the Kenneth Rainin Foundation (to RPW), the Dan and Diane Riccio Fund for Neuroscience (to RPW and PRT), F30 AI150127 (to NDP), T32 AI132152 (to NDP and JDI), T32 GM107000 (to NDP), F31 NS122423 (to JDI), and R35 GM118112 (to PRT). Some strains were provided by the *Caenorhabditis* Genetics Center, which is funded by the NIH Office of Research Infrastructure Programs (P40 OD010440).

## Additional information

### Funding

| Funder | Grant reference number | Author |
|---|---|---|
| National Institute of Allergy and Infectious Diseases | R01 AI130289 | Read Pukkila-Worley |
| National Institute of Allergy and Infectious Diseases | R01 AI159159 | Read Pukkila-Worley |
| National Institute of Allergy and Infectious Diseases | R21 AI163430 | Read Pukkila-Worley |
| Kenneth Rainin Foundation | | Read Pukkila-Worley |
| National Institute of Allergy and Infectious Diseases | F30 AI150127 | Nicholas D Peterson |
| National Institute of Allergy and Infectious Diseases | T32 AI132152 | Nicholas D Peterson Janneke D Icso |
| National Institute of General Medical Sciences | T32 GM107000 | Nicholas D Peterson |
| National Institute of Neurological Disorders and Stroke | F31 NS122423 | Janneke D Icso |
| National Institute of General Medical Sciences | R35 GM118112 | Paul R Thompson |
| The Dan and Diane Riccio Fund for Neuroscience | | Read Pukkila-Worley Paul R Thompson |

The funders had no role in study design, data collection and interpretation, or the decision to submit the work for publication.

## Author contributions
Nicholas D Peterson, Conceptualization, Data curation, Formal analysis, Investigation, Methodology, Validation, Visualization, Writing – original draft, Writing – review and editing; Janneke D Icso, Data curation, Formal analysis, Investigation, Methodology, Validation, Visualization, Writing – review and editing; J Elizabeth Salisbury, Data curation, Investigation, Validation; Tomás Rodríguez, Software; Paul R Thompson, Supervision, Writing – review and editing; Read Pukkila-Worley, Conceptualization, Data curation, Formal analysis, Funding acquisition, Investigation, Methodology, Project administration, Resources, Supervision, Validation, Visualization, Writing – original draft, Writing – review and editing

## Author ORCIDs
Nicholas D Peterson http://orcid.org/0000-0003-4157-8119
Tomás Rodríguez http://orcid.org/0000-0002-8724-5427
Paul R Thompson http://orcid.org/0000-0002-1621-3372
Read Pukkila-Worley http://orcid.org/0000-0001-5340-8294

## Decision letter and Author response
Decision letter https://doi.org/10.7554/eLife.74206.sa1
Author response https://doi.org/10.7554/eLife.74206.sa2

# Additional files

## Supplementary files
• Supplementary file 1. Genes significantly differentially expressed in uninfected *nhr-8(hd117)* mutants compared to wild-type animals in the RNA-seq experiments presented in *Figure 3D* and *Figure 3—figure supplement 1B*.

• Supplementary file 2. Genes significantly differentially expressed in uninfected *nhr-8(ok186)* mutants compared to wild-type animals in the RNA-seq experiments presented in *Figure 3—figure supplement 1A*.

• Supplementary file 3. Genes significantly differentially expressed in uninfected wild-type animals in the absence (0 μg/mL) versus presence (5 μg/mL) of cholesterol supplementation in the RNA-seq experiments presented in *Figure 3E* and *Figure 3—figure supplement 1B*.

• Supplementary file 4. Sample sizes, mean lifespan, and p values for the *C. elegans* pathogenesis assays.

• Supplementary file 5. Genes in Cluster one and Cluster 2 of the heat map shown in *Figure 6F*.

• Supplementary file 6. Primer, crRNA guide and ssODN sequences designed for this study.

• Supplementary file 7. *pmk-1* dependent genes used for GSEA analysis.

• Transparent reporting form

• Source data 1. Raw and annotated gel and blot images 2 of 2.

• Source data 2. Raw and annotated gel and blot images 1 of 2.

## Data availability
The mRNA-seq datasets are available from the NCBI Gene Expression Omnibus using the accession numbers GSE178572 and GSE190585. Source data files are provided for all figures.

The following datasets were generated:

| Author(s) | Year | Dataset title | Dataset URL | Database and Identifier |
| --- | --- | --- | --- | --- |
| Peterson ND, Pukkila-Worley R | 2021 | Sterol scarcity primes p38 immune defenses through a TIR-1/SARM1 phase transition | https://www.ncbi.nlm.nih.gov/geo/query/acc.cgi?acc=GSE178572 | NCBI Gene Expression Omnibus, GSE178572 |

*Continued on next page*

*Continued*

| Author(s) | Year | Dataset title | Dataset URL | Database and Identifier |
|---|---|---|---|---|
| Peterson ND, Pukkila-Worley R | 2021 | Pathogen infection and cholesterol deficiency activate the *C. elegans* p38 immune pathway through a TIR-1/SARM1 phase transition | https://www.ncbi.nlm.nih.gov/geo/query.acc.cgi?acc=GSE190585 | NCBI Gene Expression Omnibus, GSE190585 |

The following previously published dataset was used:

| Author(s) | Year | Dataset title | Dataset URL | Database and Identifier |
|---|---|---|---|---|
| Fletcher M, Butty V, Kim DH | 2019 | Transcriptional profiling of *C. elegans* on pathogenic *Pseudomonas aeruginosa* | https://www.ncbi.nlm.nih.gov/geo/query.acc.cgi?acc=GSE119292 | NCBI Gene Expression Omnibus, GSE119292 |

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
