## [Editor Report]

Your manuscript makes an important contribution to delineating mechanisms involved in the activation of innate immune function in *C. elegans*. The reviewers as well as the editors find this study to be well-conducted, presenting a large amount of new and convincing data on the phase transition underlying p38 immune pathway activation, especially on the novel role of cholesterol metabolism in this process.

---

## [Decision Letter]

**Decision letter after peer review:**

Thank you for submitting your article "A TIR-1/SARM1 phase transition underlies p38 immune pathway activation in the *C. elegans* intestine" for consideration by *eLife*. Your article has been reviewed by 3 peer reviewers, one of whom is a member of our Board of Reviewing Editors, and the evaluation has been overseen by Matt Kaeberlein as the Senior Editor. The following individual involved in review of your submission has agreed to reveal their identity: Emily Troemel (Reviewer #2).

Essential revisions:

1) Please add survival curves/cfu accumulation of a pmk-1;nhr-8 double mutant to the manuscript to validate the notion that the overactivation of the p38 MAPK pathway seen in the nhr-8 mutant is protective. If the double mutant is more sensitive than nhr-8, this would suggest that the p38 activation that is occurring is still protective, even if not up to wild type levels of survival. However, if the double mutant survives better than nhr-8, it suggests that the overactivation of p38 is having a detrimental effect.

2) Figure 1E: As T24B8.5p::gfp and T24B8.5 gene expression are used frequently as read-outs for p38 PMK-1 pathway activation in cholesterol deprivation assays, could the authors address why T24B8.5 is not shown as differentially expressed in the absence of cholesterol? Furthermore, why not use irg-4p::gfp (or another differentially expressed gene during cholesterol deprivation) as a reporter in main figures?

3) It is argued that knock-down of atf-7 abrogates T24B8.5::gfp activation in nhr-8 animals. However, the data in Figure 2G is not that convincing. The fluorescent phenotype appears in-between wild type and nhr-8. Can the fluorescent data be quantified or can another method be used such a qRT of the relevant genes in a nhr-8 animal with atf-7 RNAi?

4) The authors show that nhr-8 mutants have lowered *P. aeruginosa* load compared to wild-type animals and call this 'clearance'. Line 443: "promotes clearance of pathogenic bacteria from the intestine (Figure 1O)" However, to demonstrate clearance, the authors need to show more than one timepoint, demonstrating a higher level of pathogen at an earlier timepoint, and a lower level at a later timepoint. Based on previous studies with this pathogen, it is more likely that there is simply lowered accumulation of *P. aeruginosa* in nhr-8 mutants rather than active clearance. The authors either should modify their language, or do a time course to show clearance.

5) Analysis of transcriptional data: Please evaluate genes upregulated in nhr-8(ok186) and compare them to genes upregulated during infection and/or known to be associated with p38, establishing the significance of the intersection statistically.

6) Figure 5A: Would it be possible to test a solid-like compartment disrupting additive? To claim TIR undergoes a liquid-to-solid phase transition, it would be preferable to present a positive result (a solid-disrupting additive decreasing activity in support of the model) rather than the negative result that a disruption of liquid-like compartments does not sufficiently decrease NADase activity- especially as reduction in NADase activity due to liquid disruption is statistically significant.

7) The authors should better explain how the lowered *P. aeruginosa* levels in nhr-8 mutants is an example of increased pathogen resistance, which together with the lowered survival of these mutants is an example of decreased tolerance of infection. This distinction is an important aspect of pathogen infection studies, i.e. the difference between pathogen resistance (controlling levels of the pathogen) and tolerance (controlling the health impacts from the pathogen). As it stands the text is confusing, e.g. Lines 183-184: "Thus, even though nhr-8(hd117) mutant animals are more susceptible to pathogen killing, they are mounting effective anti-pathogen defenses toward an ingested pathogen".

8) Do the authors think that a lack of cholesterol is sensed by some unknown mechanism and leads to this response? Or do they favor a model in which the lack of cholesterol change the biophysical properties of the cell such that oligomerization of TIR-1 is favored? Some discussion on how the lack of cholesterol and pathogen exposure might stimulate TIR-1 oligomerization and signalling would be interesting.

9) The reviewers have questions regarding why lack of cholesterol is detrimental to survival under infection challenge, despite its effect on priming. They have suggested that additional care should be taken in discussing this result and its implication. Specific suggestions:

a. Could the authors comment on / show data on survival in the absence of infection (e.g. ageing?). Lack of cholesterol shortens nematode lifespan and has been reported to prevent benefits from caloric restriction and of some lifespan extending mutations (daf-2). What is the mechanism of such detriment? Does it make sense in the context of the proposed role in modulating innate immunity?

b. Furthermore, daf-2 mutation increases resistance to infection and this also requires p38. Given the suggested role of low cholesterol in priming nematodes for infection, how might this be reconciled? Does low cholesterol cause inappropriate / over activation of p38 which reduces survival during infection and also shortens lifespan under CR? If this is the case, how would the proposed priming be of benefit as it interferes with appropriate response to both types of challenges (lack of nutrients and infection??).

c. The authors claim that activation of p38 PMK-1 pathway in sterol-scarce environments is an "adaptive response that allows a metazoan host to anticipate environmental threats during micronutrient deprivation". However, the data could support alternative explanations for the upregulation of PMK-1 during sterol scarcity. For example, is there some other benefit to upregulating the PMK-1 pathway (in absence of sterols) that isn't related to pathogen defense? Could PMK-1 upregulation be a consequence of some other disturbance rather than a preemptive response? What other cellular processes are perturbed during cholesterol deprivation and does PMK-1 play a role in regulating these processes? The authors should consider framing the concept of an "adaptive response" in a more speculative manner and also address alternative explanations.

*Reviewer #1:*

The manuscript: "A TIR-1/SARM1 phase transition underlies p38 immune pathway activation in the *C. elegans* intestine" by Peterson et al. investigates aspects of the phase transition involved in p38 MAPK activation and innate immunity in *C. elegans* intestinal cells. The study investigates a mechanism for activation of the *C. elegans* p38 homolog, PMK-1, a required step in activation of innate immunity in *C. elegans*. The authors use a range of techniques, including fluorescence microscopy of tagged TIR-1 (Toll/interleukin-1 receptor domain protein) to demonstrate that p38 PMK activation involves physical multimerization of TIR-1 and that this assembly induces NAD+ glycohydrolase activity, which in turn is required for p38 PMK activation.

The authors further show that mutations that interfere either with physical association or with NAD+ glycohydrolase activity prevent increased expression of immune effectors and cause increase susceptibility to bacterial pathogens. A similar mechanism, involving human SARM1 and NAD+ glycohydrolase activity has previously been demonstrated to function in the context of injury and axonal degeneration in human neurons. The demonstration that phase transition of TIR-1/SARM1 and increased NAD+ glycohydrolase activity is required for activation of innate immunity in *C. elegans* intestinal cells therefore demonstrates an evolutionarily conserved mechanism functioning in different tissues and in response to different stressors.

Finally, the authors demonstrate that low dietary cholesterol activates immune effector transcription independent of bacterial infection. It is surprising that, despite cholesterol scarcity apparently promoting p38-mediated innate immunity, worms on low cholesterol / with cholesterol related mutants (nhr-8) are more rather than less susceptible to pathogen killing. The reason for this susceptibility should be further investigated or at the very least more deeply discussed. This study is unusual in that it combines very different techniques to follow the stages of the phase transition and induction of innate immune factors. The analysis of transcriptional changes in the presence and absence of cholesterol is interesting but could be taken further. Despite some limitations, I have no doubt that the insights and tools presented here will be of valuable for future studies evaluating this conserved mechanism.

Comments for the authors:

I feel that the analysis of transcriptional data could be taken further. For example, are genes upregulated in nhr-8(ok186) statistically significantly more likely to be upregulated during infection (and vice versa for downregulated genes). How does this statistic depend on known association of these genes with p38.

I am also confused why, despite priming animals for infection, lack of cholesterol is detrimental during the challenge. Would the point of such priming not be to increase survival during infection?

Could the authors comment on / show data on survival in the absence of infection (e.g. ageing?). Lack of cholesterol shortens nematode lifespan and has been reported to prevent benefits from caloric restriction and of some lifespan extending mutations (daf-2). What is the mechanism of such detriment? Does it make sense in the context of the proposed role in modulating innate immunity?

Furthermore, daf-2 mutation increases resistance to infection and this also requires p38. Given the suggested role of low cholesterol in priming nematodes for infection, how might this be reconciled? Does low cholesterol cause inappropriate / over activation of p38 which reduces survival during infection and also shortens lifespan under CR? If this is the case, how would the proposed priming be of benefit as it interferes with appropriate response to both types of challenges (lack of nutrients and infection??).

*Reviewer #2:*

The p38 MAPK stress/immune signaling pathway is conserved across a wide range of organisms. In the nematode *C. elegans*, the PMK-1 p38 pathway controls expression of anti-microbials, and is among the best studied and most important pathways for defense against pathogen infection. Despite this key role, little is known about the upstream activators of the MAPK pathway.

In this paper, the authors show that cholesterol scarcity activates the PMK-1 p38 MAPK pathway, as measured by upregulation of PMK-1 phosphorylation and increased expression of PMK-1 controlled gene expression. They also show that nhr-8 mutants, which have impaired cholesterol metabolism, have activated PMK-1 signaling. These mutants also have lowered pathogen load upon P. aeruginosa infection, albeit with impaired survival. This manuscript also shows that the SARM1 homolog TIR-1, which many studies have shown acts upstream of MAPK in *C. elegans*, forms puncta upon exposure to triggers that activate p38 MAPK pathway, such as in nhr-8 mutants that have impaired cholesterol metabolism. They also demonstrate TIR-1 puncta are induced upon pathogen infection. These puncta appear to be important for activation of PMK-1, as mutations that block TIR-1 multimerization also block PMK-1 activation. This manuscript also shows that NAD-ase activity of TIR-1 is regulated by a phase transition of TIR-1, and NAD-ase activity requires multimerization. Importantly, NAD-ase activity and multimerization of TIR-1 appear important for immune defenses in vivo, as mutations that impair these features of TIR-1 lead to increased susceptibility to infection.

Overall this manuscript makes an important contribution to the field of *C. elegans* immunity. It shows that the assembly of TIR-1 and its NAD-ase activity trigger the p38 MAPK immune pathway. It also provides insight that lowered cholesterol activates these same processes, leading to p38 MAPK signaling and pathogen resistance. Although some of these results were preceded by similar findings about TIR-1 in other systems (e.g. TIR-1 puncta were shown in *C. elegans* neurons, and the importance of TIR phase transition, multimerization and NADase activity has been shown in other organisms), there is an impressive amount of rigorous data in this manuscript, and it is a significant contribution to bring all these mechanisms together in the context of *C. elegans* intestinal immunity. However, in some cases, the claims are a bit overstated based on the data shown, with actual evidence being more indirect. There are a lot of moving parts in this paper, and the order in which information is provided can be a bit confusing.

1. The abstract states that "TIR-1/SARM1 phase transition is modified by dietary cholesterol". However, those results were not apparent in the manuscript – rather, the authors show altered TIR-1 puncta in nhr-8 mutants, which have altered cholesterol metabolism. The authors should either show that lowered cholesterol induces TIR-1 puncta, or modify their language.

Related to this issue, the authors need to provide more information about what exactly is perturbed in nhr-8 mutants, given that these mutants are used in many experiments as a proxy for cholesterol scarcity. For example, Line 126-128: How tight is the correlation between transcriptional profiles of nhr-8 loss-of-function mutants and cholesterol-deprived animals? Including a plot comparing both would further support the use of nhr-8 mutants to study conditions of low sterol content. Also, Line 153: Does increased cholesterol supplementation or cholesterol solubilization result in increased cholesterol content in worms? Quantification of cholesterol levels within animals would strengthen the claim that "immune effector activation in nhr-8(hd117) mutants is due to sterol deficiency".

2. Line 404 " these data demonstrate for the first time that physiological stressors, both pathogen and non-pathogen, induce TIR-1 multimerization into puncta within intestinal epithelial cells, which superactivates the intrinsic NADase activity of this protein complex to activate the p38 PMK-1 innate immune pathway". This sentence implies all of these findings have been shown in intestinal cells. The authors show TIR-1 puncta in intestinal cells, and show that mutants defective in NADase activity have less p38 activation, but activation of NADase activity is shown in vitro. Recommend to revise language, or to show through purification of TIR-1 from *C. elegans* intestinal cells activated by these stressors that it there is increased NADase activity.

3. A major claim in the paper mentioned above is that physiologically relevant stressors induce TIR-1 multimerization in puncta, shown in Figure 6. However, it is unclear how puncta were counted. Did the authors count puncta in a z-stack or in just one optical plane? How did the authors resolve individual puncta from overlapping puncta? Furthermore, the authors normalize puncta fluorescence to autofluorescence, but autofluorescence is altered by many conditions. Either demonstrate that autofluorescence is not affected by these triggers and genetic backgrounds, or use a different normalization method.

In addition, the authors should eliminate the concern that nhr-8 mutants are simply prone to higher levels of protein aggregation in general, e.g. by showing lack of aggregation in some unrelated protein. Showing fewer puncta in the multimerization-defective TIR-1 mutant protein would also strengthen the case that the puncta formation is due to organized TIR-1 multimerization (like the model in Figure 7L), rather than non-specific aggregation.

*Reviewer #3:*

The p38 MAPK pathway is a crucial stress-sensing pathway involved in many processes including immune activation in metazoans. It has been heavily studied in both mammalian systems as well model hosts such *Caenorhabditis elegans*. However, the mechanisms by which this pathway are activated are not yet fully understood. Using *C. elegans*, the authors demonstrate that a component of this pathway, Toll/interleukin-1 receptor domain protein (TIR-1), an NAD+ glycohydrolase homologous to mammalian sterile α and TIR motif-containing 1 (SARM1), is discovered to undergo a phase transition, forming a higher-ordered protein assembly that is necessary for activation. Strengths include robust genetic analysis as well as rigorous in vitro enzyme kinetic assys revealing that NAD+ glycohydrolase activity requires oligomerization and phase transition. This is similar to what was previously shown with mammalian SARM. What is new is that this oligomerization is visualized for the first time in live animals exposed to pathogen.

Another exciting discovery is that cholesterol limitation additionally induces oligomerization of TIR-1 and p38 MAPK pathway activation. Cholesterol, which *C. elegans* is not able to manufacture, is limited by not adding it to the medium or by preventing its adequate import using a nhr-8 mutant. nhr-8 mutants or animals raised without cholesterol are hyper-susceptible to pathogen killing. However, in this and previous work, cholesterol limitation is shown to upregulate p38 MAPK signaling. A major claim of the manuscript is that micronutrient deprivation primes *C. elegans* for pathogen defense. There are two major weaknesses with this claim. (1) It is overbroad in that cholesterol is the only micronutrient that the authors show has this effect. (2) The animals who are experiencing cholesterol deprivation survive worse on pathogen – yes, there is less pathogen accumulation and more p38 pathway activation, in fact mis/overactivation, but there is still early death. An alternative hypothesis is that the overactivation/misregulation of p38 observed in nhr-8/cholesterol-deprived animals on pathogen is maladaptive causing a detrimental effect – early death.

Comments for the authors:

1) I would like to see the survival curves/cfu accumulation of a pmk-1;nhr-8 double added to the manuscript to validate whether the authors can argue that the overactivation of the p38 MAPK pathway seen in the nhr-8 mutant is protective. If the double mutant is more sensitive than nhr-8, this would suggest that the p38 activation that is occurring is still protective, even if not up to wild type levels of survival. However, if the double mutant survives better than nhr-8, it suggests that the overactivation of p38 is having a detrimental effect.

2) Do the authors think that a lack of cholesterol is sensed by some unknown mechanism and leads to this response? Or do they favor a model in which the lack of cholesterol change the biophysical properties of the cell such that oligomerization of TIR-1 is favored? Some discussion on how the lack of cholesterol and pathogen exposure might stimulate TIR-1 oligomerization and signaling would be interesting.

3) It is argued that knock-down of atf-7 abrogates T24B8.5::gfp activation in nhr-8 animals. However, the data in Figure 2G is not that convincing. The fluorescent phenotype appears in-between wild type and nhr-8. Can the fluorescent data be quantified or can another method be used such a qRT of the relevant genes in a nhr-8 animal with atf-7 RNAi?

4) Inset in Figure 4I was not immediately clear and not described in the figure legend. It was hard to figure out what it was depicting at first. * are used to mark mutants with no activity, but there appears to be a mistake between E788Q and H833A.

[Editors' note: further revisions were suggested prior to acceptance, as described below.]

Thank you for submitting your article "Pathogen infection and cholesterol deficiency activate the *C. elegans* p38 immune pathway through a TIR-1/SARM1 phase transition" for consideration by *eLife*. Your article has been reviewed by 2 peer reviewers, and the evaluation has been overseen by a Reviewing Editor and Matt Kaeberlein as the Senior Editor. The following individual involved in review of your submission has agreed to reveal their identity: Emily R Troemel (Reviewer #2).

The reviewers and editor feel that this is an strong paper and agree that you have successfully addressed most of the earlier comments. There are still two outstanding concerns that could however be addressed either by providing additional data or by adding appropriate caveats and further discussion to the final manuscript.

Essential revisions:

1) There is still a concern that the link between cholesterol deficiency and puncta formation is indirectly shown through nhr-8(hd117); tir-1::wrmScarlet animals. This claim would be strengthened by providing confocal images of tir-1::wrmScarlet animals grown under cholesterol-deficient conditions. Otherwise, we ask the authors to reflect comment through modification of their language (e.g. "We demonstrate that a loss-of-function mutation in nhr-8, which alters cholesterol metabolism, causes TIR-1/SARM-1 to oligomerize into puncta in epithelial cells…").

2) Another remaining concern is that TIR-1 puncta formation in nhr-8 mutants may be a result of non-specific aggregation (See Original reviews). While Tergitol does suppress TIR-1::wrmScarlet puncta formation, it is possible that Tergitol would also suppress non-specific aggregation of TIR-1::wrmScarlet. Further, tir-1 RNAi data do not provide direct evidence that puncta formation is physiologically relevant. Puncta formation may be a consequence of non-specific aggregation due to dysregulation of other biological processes in nhr-8 mutants rather than a consequence of organized TIR-1 multimerization. Lack of puncta upon tir-1 RNAi treatment could reflect a reduction in TIR-1 protein levels such that non-specific aggregates are not visible. To strengthen the link between ordered multimerization and puncta formation, we ask the authors or show that multimerization mutants (i.e. tir-1∆SAM) exhibit decreased puncta or address these concerns in their interpretation

While these additional data would be beneficial, we feel that at this stage it is not essential, if appropriately addressed in the text.

Other than these two remaining points, there is consensus that you have addressed all previous concerns in full and that the manuscript makes an important contribution and is now suitable for publication in *eLife*.

---

## [Author Response]

Essential revisions:1) Please add survival curves/cfu accumulation of a pmk-1;nhr-8 double mutant to the manuscript to validate the notion that the overactivation of the p38 MAPK pathway seen in the nhr-8 mutant is protective. If the double mutant is more sensitive than nhr-8, this would suggest that the p38 activation that is occurring is still protective, even if not up to wild type levels of survival. However, if the double mutant survives better than nhr-8, it suggests that the overactivation of p38 is having a detrimental effect.

We performed three biological replicates of the suggested experiment, and these data are now presented in Figure 4M of the manuscript and discussed in the text (Lines 386-394). We also discussed these findings in the Discussion section (Lines 494-499). These experiments revealed that *tir-1(qd4);nhr-8(hd117)* double mutants were more susceptible to *P. aeruginosa* infection than *nhr-8(hd117)* mutants. As noted by this reviewer, these data suggest that the induction of the p38 PMK-1 pathway in the *nhr-8(hd117)* mutant background provides protection from *P. aeruginosa* infection. In addition, *tir-1(qd4);nhr-8(hd117)* double mutants were slightly, but significantly and reproducibly, more susceptible than the *tir-1(qd4)* mutant to killing by *P. aeruginosa*. We therefore hypothesize that the susceptibility to pathogen-mediated killing in animals that lack sufficient cholesterol [*e.g.*, the *nhr-8(hd117)* mutant] leads to the additive pathogen susceptibility in animals that also lack a functioning p38 PMK-1 host defense pathway [e.g*.*, the *tir-1(qd4)* mutant].

2) Figure 1E: As T24B8.5p::gfp and T24B8.5 gene expression are used frequently as read-outs for p38 PMK-1 pathway activation in cholesterol deprivation assays, could the authors address why T24B8.5 is not shown as differentially expressed in the absence of cholesterol? Furthermore, why not use irg-4p::gfp (or another differentially expressed gene during cholesterol deprivation) as a reporter in main figures?

For the revised version of this manuscript, we performed a new RNA-seq experiment ourselves rather than rely on publicly available data. We compared *C. elegans* exposed to 0 μg/mL cholesterol with animals exposed to 5 μg/mL cholesterol (three biological replicates each). This study revealed that T24B8.5, among other innate immune effectors, is robustly activated during cholesterol starvation. These data are now presented in the manuscript in Figure 3E. We also used qRT-PCR in three different biological replicate samples to confirm that T24B8.5 was upregulated in animals exposed to 0 μg/mL cholesterol (Figure 3C). As we observed previously, innate immune effector genes were strongly enriched during cholesterol starvation. We also found that the transcriptional changes during cholesterol starvation mirror those in the *nhr-8* loss-of-function mutants, which further supports the use of *nhr-8* mutants to study conditions of low sterol content. This comparison is presented in a new supplemental figure (Figure 3—figure supplement 1B) and discussed in the text (Lines 315-318). T24B8.5p::*gfp* is widely used as a readout of activation of the p38 PMK-1 pathway. Thus, we chose to use this reporter in the main figures.

3) It is argued that knock-down of atf-7 abrogates T24B8.5::gfp activation in nhr-8 animals. However, the data in Figure 2G is not that convincing. The fluorescent phenotype appears in-between wild type and nhr-8. Can the fluorescent data be quantified or can another method be used such a qRT of the relevant genes in a nhr-8 animal with atf-7 RNAi?

We quantified the fluorescent images in Figure 2G (Now Figure 4G) as suggested by the reviewer and present the data in a new figure (Figure 4—figure supplement 1C). Knockdown of *atf-7* by RNAi in *nhr-8(hd117)* animals significantly reduced T24B8.5p::*gfp* activation back to wild-type levels. In addition, treatment of *nhr-8(hd117)* animals with *tir-1(RNAi)* also decreased T24B8.5p::*gfp* activation.

4) The authors show that nhr-8 mutants have lowered *P. aeruginosa* load compared to wild-type animals and call this 'clearance'. Line 443: "promotes clearance of pathogenic bacteria from the intestine (Figure 1O)" However, to demonstrate clearance, the authors need to show more than one timepoint, demonstrating a higher level of pathogen at an earlier timepoint, and a lower level at a later timepoint. Based on previous studies with this pathogen, it is more likely that there is simply lowered accumulation of *P. aeruginosa* in nhr-8 mutants rather than active clearance. The authors either should modify their language, or do a time course to show clearance.

We have removed the word “clearance” from the manuscript. We now state that there is reduced accumulation of *P. aeruginosa* in the *nhr-8(hd117)* mutants, as suggested by this reviewer.

5) Analysis of transcriptional data: Please evaluate genes upregulated in nhr-8(ok186) and compare them to genes upregulated during infection and/or known to be associated with p38, establishing the significance of the intersection statistically.

In Figure 3—figure supplement 1A of the revised manuscript, we show that the genes that are upregulated during infection with *P. aeruginosa* and innate immune effector genes are strongly and significantly correlated in *nhr-8(ok186)* mutants (r=0.849 and 0.703, respectively, p<0.05 for each comparison). These data mirror the transcriptional changes in the *nhr-8(hd117)* mutant (Figure 3D).

6) Figure 5A: Would it be possible to test a solid-like compartment disrupting additive? To claim TIR undergoes a liquid-to-solid phase transition, it would be preferable to present a positive result (a solid-disrupting additive decreasing activity in support of the model) rather than the negative result that a disruption of liquid-like compartments does not sufficiently decrease NADase activity- especially as reduction in NADase activity due to liquid disruption is statistically significant.

The best-known solid disrupting reagents are detergents, which would denature the protein and lead to an artifactual loss in activity as opposed to true disruption of the solid phase. Notably, the fact that we can isolate the active fraction by centrifugation argues for a liquid-to-solid phase transition. However, given the statistical significance of the 1,6-hexanediol data, the active form of the enzyme may lie on a continuum between a liquid-to-liquid to liquid-to-solid phase transition or exist in a gel-like state. The text of the manuscript (Lines 230-234) has been updated to reflect this analysis.

7) The authors should better explain how the lowered *P. aeruginosa* levels in nhr-8 mutants is an example of increased pathogen resistance, which together with the lowered survival of these mutants is an example of decreased tolerance of infection. This distinction is an important aspect of pathogen infection studies, i.e. the difference between pathogen resistance (controlling levels of the pathogen) and tolerance (controlling the health impacts from the pathogen). As it stands the text is confusing, e.g. Lines 183-184: "Thus, even though nhr-8(hd117) mutant animals are more susceptible to pathogen killing, they are mounting effective anti-pathogen defenses toward an ingested pathogen".

We have edited this sentence, and the paragraph where it appears, to better highlight the important point raised by this reviewer. We now state: “These data suggest that low cholesterol in the *nhr-8(hd117)* mutants reduces tolerance to pathogen infection, and that the robust transcriptional induction of immune effectors in this mutant background leads to reduced accumulation of *P. aeruginosa* during infection.” (Lines 353-356) We also further discuss this observation in two new paragraphs in the Discussion section (Lines 491-506).

8) Do the authors think that a lack of cholesterol is sensed by some unknown mechanism and leads to this response? Or do they favor a model in which the lack of cholesterol change the biophysical properties of the cell such that oligomerization of TIR-1 is favored? Some discussion on how the lack of cholesterol and pathogen exposure might stimulate TIR-1 oligomerization and signalling would be interesting.

In the revised manuscript, we address this point in the Discussion section (Lines 487-490).

9) The reviewers have questions regarding why lack of cholesterol is detrimental to survival under infection challenge, despite its effect on priming. They have suggested that additional care should be taken in discussing this result and its implication. Specific suggestions:

In the revised Discussion section (Lines 491-506), we include two new paragraphs where we discuss some of the points raised below. In summary, we demonstrate that cholesterol supplementation can fully rescue the enhanced susceptibility to pathogen-mediated killing in *nhr-8* loss-of-function mutants (Figure 3M and 3N). These data suggest that cholesterol is required for pathogen tolerance in *C. elegans*; however, the mechanism behind this observation is not known. The p38 PMK-1 pathway provides protection during pathogen infection in the *nhr-8* mutant background (Figure 4M), suggesting that the enhanced susceptibility to pathogen-mediated killing in the *nhr-8* mutant is not secondary to aberrant activation of the p38 PMK-1 pathway. Cholesterol-deficient animals may therefore have general reductions in organismal fitness that cause vulnerability to pathogen infection and reduce lifespan.

a. Could the authors comment on / show data on survival in the absence of infection (e.g. ageing?). Lack of cholesterol shortens nematode lifespan and has been reported to prevent benefits from caloric restriction and of some lifespan extending mutations (daf-2). What is the mechanism of such detriment? Does it make sense in the context of the proposed role in modulating innate immunity?

The reviewer raises some very interesting points about the intersection of aging and immune regulation as it pertains to cholesterol metabolism. We chose not to discuss the effect of cholesterol on aging in this manuscript, however, so that we could keep the focus on immune regulation.

b. Furthermore, daf-2 mutation increases resistance to infection and this also requires p38. Given the suggested role of low cholesterol in priming nematodes for infection, how might this be reconciled? Does low cholesterol cause inappropriate / over activation of p38 which reduces survival during infection and also shortens lifespan under CR? If this is the case, how would the proposed priming be of benefit as it interferes with appropriate response to both types of challenges (lack of nutrients and infection??).

As discussed above in point 1, we present data from a new experiment in the revised manuscript that the p38 PMK-1 pathway provides protection during pathogen infection in the *nhr-8* mutant background (Figure 4M), which suggests that the enhanced susceptibility to pathogen-mediated killing in the *nhr-8* mutant is not secondary to aberrant activation of the p38 PMK-1 pathway.

c. The authors claim that activation of p38 PMK-1 pathway in sterol-scarce environments is an "adaptive response that allows a metazoan host to anticipate environmental threats during micronutrient deprivation". However, the data could support alternative explanations for the upregulation of PMK-1 during sterol scarcity. For example, is there some other benefit to upregulating the PMK-1 pathway (in absence of sterols) that isn't related to pathogen defense? Could PMK-1 upregulation be a consequence of some other disturbance rather than a preemptive response? What other cellular processes are perturbed during cholesterol deprivation and does PMK-1 play a role in regulating these processes? The authors should consider framing the concept of an "adaptive response" in a more speculative manner and also address alternative explanations.

In a new paragraph in the Discussion section (Lines 500-506), we present the points raised in this critique. We also edited the manuscript to frame the concept of immune priming in a low cholesterol environment in a more speculative manner, as suggested.

Reviewer #1:[…] The authors demonstrate that low dietary cholesterol activates immune effector transcription independent of bacterial infection. It is surprising that, despite cholesterol scarcity apparently promoting p38-mediated innate immunity, worms on low cholesterol / with cholesterol related mutants (nhr-8) are more rather than less susceptible to pathogen killing. The reason for this susceptibility should be further investigated or at the very least more deeply discussed.

Please see our response under point #9.

This study is unusual in that it combines very different techniques to follow the stages of the phase transition and induction of innate immune factors. The analysis of transcriptional changes in the presence and absence of cholesterol is interesting but could be taken further. Despite some limitations, I have no doubt that the insights and tools presented here will be of valuable for future studies evaluating this conserved mechanism.Comments for the authors:I feel that the analysis of transcriptional data could be taken further. For example, are genes upregulated in nhr-8(ok186) statistically significantly more likely to be upregulated during infection (and vice versa for downregulated genes). How does this statistic depend on known association of these genes with p38.

Please see our response under point #5.

Reviewer #2:[…] Overall this manuscript makes an important contribution to the field of C. elegans immunity. It shows that the assembly of TIR-1 and its NAD-ase activity trigger the p38 MAPK immune pathway. It also provides insight that lowered cholesterol activates these same processes, leading to p38 MAPK signaling and pathogen resistance. Although some of these results were preceded by similar findings about TIR-1 in other systems (e.g. TIR-1 puncta were shown in *C. elegans* neurons, and the importance of TIR phase transition, multimerization and NADase activity has been shown in other organisms), there is an impressive amount of rigorous data in this manuscript, and it is a significant contribution to bring all these mechanisms together in the context of *C. elegans* intestinal immunity. However, in some cases, the claims are a bit overstated based on the data shown, with actual evidence being more indirect. There are a lot of moving parts in this paper, and the order in which information is provided can be a bit confusing.1. The abstract states that "TIR-1/SARM1 phase transition is modified by dietary cholesterol". However, those results were not apparent in the manuscript – rather, the authors show altered TIR-1 puncta in nhr-8 mutants, which have altered cholesterol metabolism. The authors should either show that lowered cholesterol induces TIR-1 puncta, or modify their language.

In the revised abstract, we have eliminated the words “modified by dietary cholesterol.” We now state: “We demonstrate that cholesterol deficiency causes TIR-1/SARM1 to oligomerize into puncta in intestinal epithelial cells and engages its NAD^+^ glycohydrolase activity, which increases p38 PMK-1 phosphorylation, and primes immune effector induction in a manner that leads to reduced pathogen accumulation in the intestine during a subsequent infection.” (Lines 39-42)

Related to this issue, the authors need to provide more information about what exactly is perturbed in nhr-8 mutants, given that these mutants are used in many experiments as a proxy for cholesterol scarcity. For example, Line 126-128: How tight is the correlation between transcriptional profiles of nhr-8 loss-of-function mutants and cholesterol-deprived animals? Including a plot comparing both would further support the use of nhr-8 mutants to study conditions of low sterol content. Also, Line 153: Does increased cholesterol supplementation or cholesterol solubilization result in increased cholesterol content in worms? Quantification of cholesterol levels within animals would strengthen the claim that "immune effector activation in nhr-8(hd117) mutants is due to sterol deficiency".

We performed the experiment suggested by the reviewer and present the data in the revised manuscript in a new figure (Figure 3—figure supplement 1B) and in the text (Lines 315-320). We compared the transcriptional changes in the *nhr-8(hd117)* mutants with the genes that are induced in wild-type animals during cholesterol starvation and found a remarkable correlation between these datasets across all genes (r = 0.578) and differentially expressed genes (r = 0.836). Together with previous work that characterized the role of *nhr-8* in cholesterol homeostasis in *C. elegans* (Cell Metab 2013;18:212), these data support the use of *nhr-8* mutants to study conditions of low sterol content.

2. Line 404 " these data demonstrate for the first time that physiological stressors, both pathogen and non-pathogen, induce TIR-1 multimerization into puncta within intestinal epithelial cells, which superactivates the intrinsic NADase activity of this protein complex to activate the p38 PMK-1 innate immune pathway". This sentence implies all of these findings have been shown in intestinal cells. The authors show TIR-1 puncta in intestinal cells, and show that mutants defective in NADase activity have less p38 activation, but activation of NADase activity is shown in vitro. Recommend to revise language, or to show through purification of TIR-1 from *C. elegans* intestinal cells activated by these stressors that it there is increased NADase activity.

We edited this sentence as follows: “In summary, the above data demonstrate for the first time that physiological stresses, both pathogen and non-pathogen, induce TIR-1 multimerization into puncta within intestinal epithelial cells, which then activates the p38 PMK-1 innate immune pathway through the intrinsic NADase activity of the TIR-1 protein complex.” (Lines 413-416)

3. A major claim in the paper mentioned above is that physiologically relevant stressors induce TIR-1 multimerization in puncta, shown in Figure 6. However, it is unclear how puncta were counted. Did the authors count puncta in a z-stack or in just one optical plane? How did the authors resolve individual puncta from overlapping puncta? Furthermore, the authors normalize puncta fluorescence to autofluorescence, but autofluorescence is altered by many conditions. Either demonstrate that autofluorescence is not affected by these triggers and genetic backgrounds, or use a different normalization method.

In the revised manuscript, we devised a different method of quantifying TIR-1::wrmScarlet that was not subject to potential bias from differences in autofluorescence between experimental and control conditions. We present the details of this approach in a new section of the Materials and methods titled “Microscopy image analyses” (Lines 690-700). In summary, Fiji Image Analysis Software was used to identify TIR-1::wrmScarlet puncta in an automated fashion in the red channel. Importantly, we used a stringent cut-off for puncta calling to maximize specificity. We then overlayed the location of the red channel puncta in the green channel and determined if there were autofluorescent puncta in this exact location. TIR-1::wrmScarlet puncta were those that were present in the red, but not the green, channel. Importantly, we only reported puncta that were positively identified by the image analysis software. The result of this approach was a rigorous assessment of the difference in puncta between the experimental conditions, but one that, by design, under-called puncta numbers in the induced conditions (*i.e.,* during *P. aeruginosa* infection or in the *nhr-8* mutant). For example, occasionally, the image analysis software identified overlapping puncta as a single punctum. In this case, we reported the number as a single punctum rather than manually updating the count.

In addition, the authors should eliminate the concern that nhr-8 mutants are simply prone to higher levels of protein aggregation in general, e.g. by showing lack of aggregation in some unrelated protein. Showing fewer puncta in the multimerization-defective TIR-1 mutant protein would also strengthen the case that the puncta formation is due to organized TIR-1 multimerization (like the model in Figure 7L), rather than non-specific aggregation.

Two key observations demonstrate that the organization of TIR-1::wrmScarlet in *nhr-8* mutants is physiologically relevant and not secondary to non-specific protein aggregation. First, Tergitol, which solubilizes cholesterol and fully suppresses immune activation in *nhr-8* mutants, also suppresses TIR1::wrmScarlet puncta formation. Second, *tir-1* RNAi ablates TIR-1::wrmScarlet puncta and also suppresses immune activation.

Reviewer #3:[…] A major claim of the manuscript is that micronutrient deprivation primes *C. elegans* for pathogen defense. There are two major weaknesses with this claim. (1) It is overbroad in that cholesterol is the only micronutrient that the authors show has this effect.

We removed the words “micronutrient deficiency” from the manuscript. We now use “cholesterol deficiency.”

(2) The animals who are experiencing cholesterol deprivation survive worse on pathogen – yes, there is less pathogen accumulation and more p38 pathway activation, in fact mis/overactivation, but there is still early death. An alternative hypothesis is that the overactivation/misregulation of p38 observed in nhr-8/cholesterol-deprived animals on pathogen is maladaptive causing a detrimental effect – early death.

Please see our response under point #1.

[Editors' note: further revisions were suggested prior to acceptance, as described below.]

Essential revisions:1) There is still a concern that the link between cholesterol deficiency and puncta formation is indirectly shown through nhr8(hd117); tir-1::wrmScarlet animals. This claim would be strengthened by providing confocal images of tir-1::wrmScarlet animals grown under cholesterol-deficient conditions. Otherwise, we ask the authors to reflect comment through modification of their language (e.g. "We demonstrate that a loss-of-function mutation in nhr-8, which alters cholesterol metabolism, causes TIR-1/SARM-1 to oligomerize into puncta in epithelial cells…").

We have edited the sentence in the abstract as suggested by this reviewer (lines 39-41). This sentence is: “Finally, we demonstrate that a loss-of-function mutation in *nhr-8*, which alters cholesterol metabolism and is used to study conditions of sterol deficiency, causes TIR-1/SARM1 to oligomerize into puncta in intestinal epithelial cells.

We also made similar edits to the Introduction (lines 89-90) and Discussion (476-477).

2) Another remaining concern is that TIR-1 puncta formation in nhr-8 mutants may be a result of non-specific aggregation (See Original reviews). While Tergitol does suppress TIR-1::wrmScarlet puncta formation, it is possible that Tergitol would also suppress non-specific aggregation of TIR-1::wrmScarlet. Further, tir-1 RNAi data do not provide direct evidence that puncta formation is physiologically relevant. Puncta formation may be a consequence of non-specific aggregation due to dysregulation of other biological processes in nhr-8 mutants rather than a consequence of organized TIR-1 multimerization. Lack of puncta upon tir-1 RNAi treatment could reflect a reduction in TIR-1 protein levels such that non-specific aggregates are not visible. To strengthen the link between ordered multimerization and puncta formation, we ask the authors or show that multimerization mutants (i.e. tir-1∆SAM) exhibit decreased puncta or address these concerns in their interpretation.

We added a paragraph to the manuscript to acknowledge the possibility that the TIR-1 puncta formation in *nhr-8* mutants may be a result of non-specific protein aggregation and to point out that the in vitro and in vivo data presented in this manuscript, when considered together, suggest that this is not the case (lines 416-427). This paragraph is: “It is possible that the organization of TIR-1::wrmScarlet into visible puncta in *nhr-8* mutants is secondary to non-specific protein aggregation; however, the in vitro and in vivo data presented in this manuscript, when considered together, suggest that this is not the case. Organized multimerization of TIR-1 is a prerequisite for the NADase activity of the protein complex in vitro (Figure 2). Accordingly, the mutations that specifically abrogate TIR-1 multimerization in vitro (Figures 2J, 2K, Figure 2—figure supplement 1I), also block p38 PMK-1 pathway activation (Figures 1E and 1F) and prevented immune effector induction in response to cholesterol deprivation in vivo (Figures 5C-F). Furthermore, Tergitol, which solubilizes cholesterol and fully suppresses p38 PMK-1 immune activation in *nhr-8* mutants (Figure 5G-5N), also suppresses TIR-1::wrmScarlet puncta formation (Figure 5A and 5B). Finally and perhaps most importantly, *P. aeruginosa* infection, a separate physiological stress, also induces TIR-1::wrmScarlet puncta formation to activate the p38 PMK-1 pathway (Figure 1).”